# Constraining the relationships between aerosol height, aerosol optical depth and total column trace gas measurements using remote sensing and models

Shuo Wang[1], Jason Blake Cohen[1, 2, *], Chuyong Lin[1], Weizhi Deng[1]

[1]School of Atmospheric Sciences, Sun Yat-Sen University, Guangzhou, 519000, China
[2]Southern Marine Science and Engineering Guangdong Laboratory (Zhuhai)

*Correspondence to*: Jason Blake Cohen (jasonbc@alum.mit.edu)

**Abstract.** Proper quantification of the aerosol vertical height is essential to constrain the atmospheric distribution and lifetime of aerosols, as well as their impact on the environment. We use globally distributed, daily averaged measurements of aerosol stereo heights of fire aerosols from MISR to understand the aerosol distribution. We also connect these results with a simple plume rise model and a new multi-linear regression model approach based on daily measurements of $NO_2$ from OMI and CO from MOPITT to understand and model the global aerosol vertical height profile over biomass burning regions. First, plumes associated with the local dry-burning season at mid to high latitudes frequently have a substantial fraction lofted into the free troposphere, and in some cases even the stratosphere. Second, plumes mainly associated with less polluted regions in developing countries and heavily forested areas tend to stay closer to the ground, although they are not always uniformly distributed throughout the boundary layer. Third, plumes associated with more serious loadings of pollution (such as in Africa, Southeast Asia and Northeast China) tend to have a substantial amount of smoke transported uniformly through the planetary boundary layer and up to around 3 km. Fourth, the regression model approach yields a better ability to reproduce the measured heights as compared to the plume rise model approach. This improvement is based on a removal of the negative bias observed from the plume model approach, as well as a better ability to work under more heavily polluted conditions. However, over many regions, both approaches fail, requiring deeper work to understand the physical, chemical, and dynamical reasons underlying the failure over these regions.

## 1 Introduction

Over the past few decades, there has been an increasing amount of research into the spatial and temporal distribution of atmospheric aerosols (Achtemeier et al., 2011; Cohen et al., 2017; Cohen et al., 2018). This has been in part because of the impacts that aerosols have on clouds, radiation, the atmospheric energy balance and climate, human health, and ecosystems, among other aspects (Cohen, 2014; Tao et al., 2013; Ramanathan et al. 2007; Ming Y. et al, 2010). However, there has not been a significant amount of research work done in terms of understanding the vertical distribution of aerosols in the atmosphere (Cohen et al., 2018), although such knowledge is essential to constrain their impacts the atmospheric energy budget (Kim et al., 2008; Grandey et al., 2018), circulation, clouds and precipitation(Cohen et al., 2011; Tosca et al., 2011; Singh et

al., 2018), and ultimate tropospheric distribution (Leung et al., 2007; Randles, et al., 2017). Large-scale reviews of the biomass burning literature spend a lot of time on how the atmosphere impacts the burning conditions, but also tend to overlook the issue of how the emissions are rapidly vertically distributed upon being emitted (Palacios-Orueta et al., 2005).

The vertical distribution of aerosols is observed to be more complex than the present generation of global and mesoscale models are capable of reproducing in regions where there are multiple sources with similar magnitudes and very different vertical lofting properties (Kahn et al., 2008; Petrenko et al., 2012; Chew et al., 2013). While on one hand urban sources are emitted with relatively low amounts of heat and are therefore known to remain in the boundary layer (Guo et al., 2016), on the other hand, biomass burning sources are emitted with large amounts of heat at high temperature and frequently are rapidly transported higher in the atmosphere, such that they are effectively emitted at height (Ichoku et al., 2008; Field et al., 2009; Freeborn et al., 2014). Furthermore, there are other forcing mechanisms, such as deep convection (Petersen and Rutledge 2001; Turquety et al., 2007), volcanos (Singh et al., 2018; Flower and Kahn, 2017), mountain slope winds (Cohen et al., 2017), and other dynamical forcings (Cohen et al., 2011; Tosca et al., 2011) which also have a substantial effect on the vertical distribution of aerosols over specific spatial and temporal scales. The vertical distribution of aerosols has a direct impact on their lifetime and hence atmospheric loading, with aerosols lofted above the boundary layer having a significantly larger impact the atmosphere than those emitted into the boundary layer (Nelson et al., 2013; Paugam et al., 2016). Therefore, understanding the vertical distribution over the source regions (Nelson et al., 2013) of aerosols and how this may change over time is absolutely critical for our being able to better constrain the environmental and atmospheric impacts.

Currently, aerosol data comes from different measurements made from the surface, balloons, aircraft, and satellites, with varying degrees of accuracy (Husar et al., 1997, Jost et al., 2004, Rogers et al., 2011) used in-situ measurements to observe the plume from North American fires emitted at a surface temperature above 380K, and found that carbon monoxide and tiny particles were detected in the stratosphere at an altitude of 15.8 km. Kahn et al. (2007) found using MISR measurements that 5% to 18% of smoke plumes reached the free troposphere over Alaska and the Yukon Territories in 2004. Winker et al. 2013 introduced the idea of possibly using CALIPSO lidar as a measurement technique, since it is more sensitive to dispersed vertical aerosols away from fire points than MISR satellites, and therefore could capture the overall smoke cloud better. Val Martin et al. (2018) used MISR data with pixel-weighted and AOD-weighted statistics to estimate the impact of fire severity of on fire height and found that while in the Arctic there were significant areas with aerosols found above the boundary layer, in agricultural areas and most other non-arctic areas, the amount was small or non-existent. Cohen et al. (2018) produced the first comprehensive study using CALIPSO lidar data anywhere in the world, and found that throughout the 2006 biomass burning season in Southeast Asia that 51% to 91% of smoke from fires was ultimately found to reside in the free troposphere. This is consistent with earlier theory by which show that when a plume is injected into the free troposphere, it tends to accumulate in a relatively stable layer (Val Martin et al., 2010; Kahn et al., 2007).

The present generation of models have not been found to reproduce the vertical distribution of aerosols very well (Val Martin et al., 2012; Paugam 2016; Cohen et al., 2018). Most of the previous approaches to simulate convection induced by a fire or other surface heat sources have been performed with simplified models (Briggs, 1965; Trentmann et al., 2006). There have been multiple studies using global and regional chemical transport models (CTMs) with such simple plume models built in to try to understand the impact of fire emissions on air quality and atmospheric composition (Pfister et al., 2008; Turquety et al., 2007; Spracklen et al., 2009; Ichoku and Ellison 2014). There have also been other attempts to simulate the impacts of different vertical distributions based on higher-resolution wind patterns profiles, done on a region-by-region basis (Cohen and Prinn, 2011; Cohen and Wang, 2014). More recently, people have attempted to use Lagrangian models such as Dewitt and Gasore (2019) and Vernon et al. (2018), to understand how knowledge of air mass flows could better contribute to the understanding of different vertical regions having material from biomass burning plumes found far upwind. Val Martin et al. (2012) used a 1-D plume rise model to study plume heights over North America, which demonstrated dynamical heat flux and atmospheric stability structure affect plume rise. Cohen et al. (2018) also adapted a plume rise model and found that significant enhancements were required to the measured Fire Radiative Power (FRP) values in order to match the mean values of measured heights, although the upper and lower quartiles were not able to be successfully reproduced. At present, there is no known modelling work that can accurately and consistently reproduce this substantial atmospheric loading found throughout different regions of the world in the upper boundary layer and free troposphere.

Biomass combustion is a major source of trace gases and aerosols in the atmosphere as well as having an important impact on tropospheric ozone formation. The vast majority of biomass burning in the tropics and non-tropical agricultural regions of the world is a man-made activity (Kauffman et al., 2003; Achtemeier et al., 2011; Paugam et al., 2016), while in certain arctic regions, lighting accounts for a significant amount of biomass burning (Generoso et al., 2007). In particular, this activity has been shown to have a strong annual cycle (Cohen et al., 2017; Labonne et al., 2007; Tsigaridis et al., 2014). The process of burning releases heat, increasing the local temperature of the surrounding air, resulting in a change in buoyancy and an ensuing updraft above the heat-producing area. Based on how long the plume maintains its buoyancy, it will rise to a fairly high position in the atmosphere. However, strong turbulence causes the plume to mix with the surrounding air, reducing plume temperature and buoyancy, eventually reaching a stable layer at which the updraft stops (Damoah et al., 2006; Freitas et al., 2007). For these reasons, a significant amount of the material emitted from biomass combustion is lofted above the surface, as compared with urban sources, where almost all of the aerosol remains near the surface (Ichoku et al., 2008; Cohen and Prinn 2011). This point is important because if aerosol is injected into the atmosphere above the planetary boundary layer (PBL) they can be carried by the faster free tropospheric winds farther away, leading to a larger impact on the atmosphere (Vernon et al., 2018; Nelson et al., 2013).

The present generation of models has difficulty to reproduce the actual vertical distribution of atmospheric aerosols when addressing cases that do not tend to have a combination of a highly energetic fire source, a relatively dry atmosphere, and a

relatively optically thin smoke column emitted by the fire. One reason stems from the fact that in-situ production and removal mechanisms of aerosols as well as the distribution of rainfall are not fully understood (Tao et al., 2012), all of which weaken the ability of simple plume rise models to reproduce the vertical distribution of aerosols (Urbanski 2014; Cohen et al., 2017). In addition, heterogeneous aerosol processing associated with the highly polluted conditions within the atmospheric plume may also change the hygroscopicity, which in turn impacts the washout rate and vertical distribution of the aerosols (Kim et al., 2008; Cohen et al., 2011). On top of this, highly polluted aerosol loadings, especially so for absorbing aerosols as found in fires, lead to changes in the radiative equations and the vertical atmospheric stability (Guo et al., 2019; Cohen et al., 2018; Zhu et al., 2018). Furthermore, small scale convective events and large-scale circulation patterns are generally not both well produced by the same scale models, leading to an inherent bias against one or the other convection producing source (Winker et al., 2013; Jost et al., 2004). In summary these factors can lead to actual changes in the vertical distribution of aerosols that simple models are not able to reproduce, including those which have used inverse modeling with a fixed vertical a priori (i.e. Heald et al. 2004; Cohen and Prinn, 2011), in turn affecting the distribution of aerosols hundreds to thousands of kilometers downwind, supporting new measurement-based perspectives (i.e. Kahn 2020).

This work describes a new approach to comprehensively understand global-scale, daily measurements of the vertical distribution of aerosols, and introduces a simple modeling approach better capable of reproducing the vertical distribution of smoke aerosols emitted by biomass burning. First, we analyze the plume heights from three and a half years of daily Multi-angle Imaging SpectroRadiometer (MISR) satellite measurements, separating the more than 67,000 measurements by the magnitude of the measured variability. Next we build aerosol plume injection models depending on the region, terrain, land type, and geospatial properties. We use this simple plume model to show that the aerosol injection heights are underestimated. We then apply a linear statistical model and show that including measurements of column gas loadings from other satellites in combination with the meteorological and FRP measurements produces a better match. We imply that ignoring the magnitude of the source emissions is an important factor in the plume rise height, another factor which the current generation of models does not take into consideration. We also demonstrate that improvements in the local convective transport process and direct and semi-direct effects of aerosols are needed to further reduce the error between the models and measurements.

It is hoped that these results will provide insights to further improve our understanding of the vertical distribution of aerosols, both from the modeling side, and from what sources of information are best required from the measurement community to help the modelers improve their understanding. We also provide a unique perspective on the connections between air quality and the vertical distribution of particulate matter, allowing the community make further advances in these fields as well as associated issues of long-range transport of aerosols as well.

## 2 Methodology

### 2.1 MISR Aerosol Height Measurements

MISR, the Multi-angle Imaging SpectroRadiometer, is an instrument flying on the Terra satellite capable of recording images at 9 different angles over 4 bands at 446nm, 558nm, 672nm, and 866 nm. The cameras point forward, downward, and aftward, allowing images to be acquired with nominal view angles, relative to the surface of 0, ±26.1, ±45.6, ±60.0, and ±70.5 degrees. All cameras have a track width of 360 km and observations extending within ±81 degrees latitude (Kahn et al., 2007). In this paper, we use the MISR INteractive eXplorer (MINX) software, which captures the plume height from the MISR image

and combines it with the MODIS fire point measurements (also taken on the Terra satellite). The software then calculates the wind speed and the elevation of contrast elements globally over a 1.1 km pixel area, providing a digitization of wildfire smoke plume height (Val Martin et al., 2010; Kahn et al., 2007; Nelson et al., 2013).

### 2.2 Geography

Around the world, biomass burning and deforestation have undergone tremendous changes in the past few decades, with

current extremes making the news in many places throughout the world. To better interpret the land use conditions in the biomass burning areas, we apply global land-cover type data of 18 different vegetation types, as measured in 2015 in Fig. 1. This work specifically focuses on those areas where the land type has undergone known significant changes from forest to agriculture, or from forest or agriculture to urban, as demonstrated in the black boxes in Fig. 1.

Considering MISR daily plume heights (where the 1.1km pixels are first averaged to 10km x 10km grids) throughout the

globe, we have determined that the respective average and standard deviation of the plume height over the three and a half years of MISR daily measurements (from January 1 2008 through June 30 2011) are 1.37km and 0.72km. However, over our regions of interest, we find that we are able to capture the large bulk of the standard deviation globally, as demonstrated in Table S1.

The geographical data yields us a few conclusions about those regions which have the largest contribution to the biomass

burning height variation. First, they are distributed in the middle and low latitudes (between the Tropic of Cancer and the Tropic of Capricorn) and/or high latitudes (near the Arctic Circle). Second, they tend to occur in regions of more rapid economic growth, and/or in regions which are experiencing the most rapid change in land surface temperature.

### 2.3 MOPITT Carbon Monoxide (CO) Measurements

Carbon monoxide (CO) is a colorless and odorless gas that plays a major role in moderating the chemistry of the Earth's

atmosphere as well as having a deleterious effect on human health. One of the world's major sources of CO is emissions from biomass burning (Lin et al., 2020b). For these reasons, we obtain measurements of CO from the MOPITT satellite (an

instrument mounted on NASA's Terra satellite), which has collected data since March 2000. MOPITT's resolution is 22 km at nadir and observes the Earth in swaths that are 640 km wide.

In terms of the CO from MOPITT, we take the day time only retrievals (to reduce bias) from version 8, level 3 data, from January 1, 2008 through June 30, 2011. In specific we use the combined thermal and near infrared product (Deeter et al, 2017). We further constrain the data to where the cloud fraction is less than 0.3 and where the vertical degrees of freedom are larger than 1.5. This combination has been shown to allow us to trust that there is a sufficient amount of signal and knowledge to demonstrate an actual measurement in the vertical, as compared with a result only dependent on the a priori model, as shown in Lin et al. (2020a). There are further gaps in the data due to orbital locations and very high aerosol conditions, all of which prevent entire coverage of our areas of interest each day. Therefore, we average all of the individual MOPITT data that passes our test to a 1°x1° grid.

## 2.4 OMI Nitrogen dioxide (NO$_2$) Measurements

Another chemical species co-emitted by biomass burning with aerosols and CO is NO$_2$ (Seinfeld and Pandis, 2006). For this reason, we also use the daily average total column loading of NO$_2$ as measured by the Ozone Monitoring Instrument (OMI). In specific we use version 3 Level 2 measurements taken from the Aura satellite (Boersma et al., 2007; Lamsal et al., 2011; Levelt et al., 2006), which has ground pixel sizes ranging from 13kmx24km at nadir to about 13kmx150km at the outermost part of the swath. In terms of the NO$_2$ from OMI, we first take the daily retrievals under the conditions where the cloud fraction is less than 0.3. Next, we aggregate the data to 0.25° x 0.25° using a linear interpolation and area weighted approach. In this way, those measurements near the edge of the swath or which are adjacent to cloudy areas are weighted less heavily in terms of the merged product. However, the areas are sufficiently large as to be roughly representative of the emissions from biomass burning of the NO$_2$ from within the grid box, as compared to that transferred from adjacent grid boxes.

One advantage of the OMI NO$_2$ column measurements is that they can often be observed under relatively cloudy or smoky conditions (Lin et al., 2014). Another advantage is that the atmospheric lifetime of NO$_2$ is only a few hours, and therefore relatively large changes in the temporal-spatial distribution of NO$_2$ column measurements is highly correlated with wildfire sources (Lin et al. 2020a). NO$_2$ has another interesting property in that its production/emissions is a strong function of the temperature at which the fires are burning, since NO$_2$ is formed based on the air temperature (Seinfeld and Pandis, 2006).

## 2.5 Plume Rise Model

Although emissions from biomass burning are similar to those from urban combustion sources, there are two major differences, arising from the much higher burning temperature and the environment in which the combustion occurred. This ensures that a significant amount of the emissions from biomass burning will be transported upwards due to the positive buoyancy generated by the fire. Due to the confluence of both local and non-local dynamical forcing in-situ, the ultimate height

reached by these emissions is a complex function of the local fire energy and both the local and large-scale meteorology at the time of combustion. While the aerosol particles are immediately transported horizontally by the large-scale winds, their vertical rise will only stop once their local buoyancy has reached equilibrium, and any dynamical motion has degraded back to the background conditions (Freitas et al., 2007; Sofiev et al., 2012; Val Martin et al., 2018).

To approximate this rise, we use a simple plume rise model (Briggs, 1965) to generate the final injection height of the biomass burning emissions based on the buoyancy and horizontal velocity of the plume and various atmospheric conditions. Although this model is based on an empirical formula mathematically, it is essentially a thermodynamic approximation (Cohen et al., 2018) which costs much less computationally as well as being quite efficient when the biomass burning source covers a large area.

In theory, if such an approach was successful, and it was given appropriate environmental data, it should be able to reproduce the heights derived from the MISR multi-angle measurements. For this reason, we use a 1-D plume rise model to independently predict the position and height of each measured MISR plume at each 10km x10km grid which is found to have measurements. To initialize the model, we require meteorological data as well as MODIS hot-spot information.

## 2.6 NCEP Reanalysis Data

NCEP and NCAR produce an analysis/prediction system to produce a meteorological field analysis of the 6-hourly state of the atmosphere from 1948 to the present. The measurements incorporated into this approach include ground based, in-situ, and remotely sensed sources, while the modeling aspect is based on state-of-the-art meteorological models. Daily data for each day which has MISR data is obtained from the reanalysis version 1 (Kalnay et al., 1996). Only data required to run the plume rise model is used: the vertical temperature and pressure distributions, the surface air temperature, and the initial vertical velocity of the smoke emissions (dP/dt). The vertical temperature gradient (dT/dz) and the vertical velocity (dz/dt) are computed from this data.

## 2.7 Regression Model

Linear regression is a simple method by which one can relate the impact that a set of orthogonal inputs have in terms or reproducing measured environmental values. It does not imply causation, merely demonstrating that the input values behave in a similar manner. However, when looking to describe whether or not a new variable has a substantial amount of correlation with a given phenomenon, it can be found to be very useful.

In this case, we are interested to see if the loadings of $NO_2$ and CO are related to the heights of the fires. There is a strong physical case to be made here, since both are directly emitted by the fires themselves. Furthermore, the underlying causes of these substances are different: $NO_2$ is a function of the fire temperature, while CO is a function of the Oxygen availability. Furthermore, these are proxies for radiatively active substances such as soot and ozone.

For our work, we have decided to apply a simple linear regression model of the wind speed, FRP, CO, $NO_2$, and the ratio of $NO_2$/CO. FRP is the measure of the radiative energy released by the fire. It is usually found in the infrared part of the spectrum as this is the part of the EM spectrum that corresponds closely with the temperatures that fires occur at in the Earth System. This is because the traditional plume rise models always include wind speed and FRP in their representations, so we wanted to specifically include as many different representations of the co-emitted gasses as well, as given in Equations 1-7.

$$H_1 = \alpha * V_{wind} + \beta * W_{FRP} + \gamma * [CO] + \delta * [NO_2] + C \ , \tag{1}$$

$$H_2 = \alpha * V_{wind} + \beta * W_{FRP} + \gamma * [CO] + \varepsilon * ([NO_2]/[CO]) + C \ , \tag{2}$$

$$H_3 = \alpha * V_{wind} + \beta * W_{FRP} + + \delta * [NO_2] + \varepsilon * ([NO_2]/[CO]) + C \ , \tag{3}$$

$$H_4 = \alpha * V_{wind} + \beta * W_{FRP} + \gamma * [NO_2] + s \ , \tag{4}$$

$$H_5 = \alpha * V_{wind} + \beta * W_{FRP} + \gamma * [CO] + C \ , \tag{5}$$

$$H_6 = \alpha * V_{wind} + \beta * W_{FRP} + + \varepsilon * ([NO_2]/[CO]) + C \ , \tag{6}$$

$$H_7 = \alpha * V_{wind} + \beta * W_{FRP} + C \ , \tag{7}$$

We calculate all of the correlation coefficients ($R^2>0.2$) between the different models and the MISR measurements, ensuring that ($P<0.05$). Finally, we analyze both the magnitude of the regression coefficient as well as the magnitude of the various best-fit terms. These models are then used to reproduce the aerosol heights and are ultimately compared with both the plume model and the actual measurements.

The seven different regression models were chosen so as to cover the entire combination of different ways to fairly and uniformly incorporate the CO and $NO_2$ measurements as well as their underlying physical meanings. The 7th regression model is the approximation of the plume rise model. The 4th and 5th regression models are the approximations of the single-species linear impact of $NO_2$ and CO respectively. The 6th regression model approximates the single-species non-linear impact of $NO_2$ and CO in tandem. Finally, the 1st through 3rd regression model are the approximations of the combination of CO and $NO_2$ in tandem with both linear (model 1), or with one linear and one non-linear combination (models 2 and 3). This approach is consistent with and follows from some of the earlier works which tries to use advanced learning to understand some higher order, simple non-linear forcings, still based on some physical consideration, i.e. Cohen and Prinn, 2011.

**2.8 MERRA**

To obtain another independent dataset of aerosol height over the biomass burning regions, we use the NASA MERRA-2 hydrophobic black carbon product (Randles, et al., 2017). MERRA is a reanalysis product based on the GEOS-5 GCM and meteorology suite with an output resolution of $0.5°$ x $0.625°$ every 3 hours. The underlying aerosol model is based on GOCART aerosol, which assumes independent, non-mixed aerosols, and hence is not an ideal environment for the high concentrations

and intense mixing that occurs over biomass burning regions (Petrenko et al., 2012). The assimilated aerosol fields are mostly from MODIS and AVHRR, with a small amount of input from MISR over bright surfaces and AERONET where it exists. There are however known issues with respect to MERRA and biomass burning (i.e. Buchard et al., 2015). For these reasons, we average the 8 3-hour time periods together for each respective day of interest, and use the information from 500mb to the surface.

## 3 Discussion and Results

We approach this problem with additional measurements compared to what are normally made so that we can have a deeper insight into how these somewhat related species have on height to which aerosols from biomass burning rise in the atmosphere. Due to the fact that there are additional processes in-situ which can lead to heating, cooling, and other changes to the dynamics, it is essential that we establish any first-effect relationships, and then work more deeply as a community to address them in turn.

First, we to enforce consistency, we impose a condition that for all days analyzed, we must have data present from all of the data sources: MISR, MODIS, MOPITT and OMI. On this basis, we explore the relationships between the two basic data sets (MISR and MODIS) and the source regions. By choosing regression models that both represent the format of the plume rise model as well as those that do not, but are instead based on additional information from MOPITT and OMI, we are thereby including this data in a way that is consistent with the underlying science and without bias.

Second, since these datasets make measurements with different assumptions, we also will reduce our bias in our inputs measurements as a function of clouds, different burning conditions, radiation feedbacks, and other actual atmospheric effects. We hope that this will help us to more clearly clarify the actual atmospheric phenomena responsible for the vertical transport, which a more conventional plume rise model may not be able to account for.

Third, the range of the seven regression models is an attempt to intelligently account for the fact that the column loadings of the CO and $NO_2$ offer physical meaning and insight, as compared to merely being an attempt to minimize any unexplained variance. We argue that the column values of both CO and $NO_2$ are both directly and indirectly related to the magnitude and the height of the vertical aerosol column. Due to the fact that the emissions of $NO_2$ is a strong function of the fire temperature, and its short atmospheric lifetime, the $NO_2$ is strongly related to the temperature of the fire, or the FRP, which is one of the essential driving forces of the buoyancy. This issue is strongly coupled with the fact that FRP is also one of the most error-prone of the measurements commonly used to drive the plume-rise models, with the FRP commonly underestimated in the tropics due to clouds and aerosols, as given in Kaiser et al. (2012), Cohen et al. (2018), and Lin et al. (2020a). Additionally, the amount of CO produced is a function of the total amount of biomass burned as well as the wetness of the surface itself where the burning occurred, and hence the CO column loading is also physically related to the properties of the fires. In fact,

using a measure of the CO column can help us to overcome the physical constraints that current measurements have in terms of addressing the issues of how much peat or understory has burned, or if such fires which are occurring without direct line of sight from above can even be detected by the current fire detection processes at all (Leung et al., 2007; Ichoku et al., 2008). The combination of high $NO_2$ (which is more produced at higher temperature) and low CO (which is more produced at higher temperature) means that the ratio of $NO_2$ to CO also provides further physical insight into the non-linearities associated with the fire temperature, wetness, and possibility of other heat sources/sinks at the fire/atmosphere interface such as smoldering, conversion to latent heat, etc.

**3.1 Characteristics of MISR, OMI and MOPITT species**

We use a PDF analysis to look at the distribution of the daily fire-constrained aggregated Measurements from MISR from each region over the entire dataset from 2008 to 2011 in Fig. 2. The statistical mean and standard deviation over each region are given in Table S1. We determine that the height of measurements ranges from 0 to 29 km (with extremely high values in the middle stratosphere possibly an error), which is not only higher than previous studies (Cohen et al., 2018; Val Martin et al., 2018), but also includes some extreme events which have made their way into the stratosphere. Due to the fact that first, the majority of the plumes are injected into the boundary layer or the lower free troposphere, second that this paper is not looking into the underlying physics of stratospheric injection (Pengfei Yu et al., 2019), and third that plumes tend to accumulate within layers of relative atmospheric stability, therefore an upper bound cutoff on the measured values of 5000m is imposed. This is consistent with the fact that over the regions of interest in this work, fewer than 0.48% of the total plume heights are more than 5 km.

There are very different distributions of the measured heights over the different regions Fig. 2. The corresponding mean, standard deviation, and skewness of the heights over each respective region is given in Table S1. The average percentage of the data which has a measured height above 2 km (selected because it is always in the free troposphere) is 15.0%, with the lowest in Central Canada of 41.7% and the highest in Midwest Africa of 0.8%. In terms of the amount of data measured with a height more than 4km, the average over the globe is 1.5%, while the range is as high as 6.6% in Central Canada and as low as 0.1% in Midwest Africa and Northern Australia. On the other end of the comparison, some regions which are very polluted near the surface, while others show the vast majority of their heights are elevated off the ground. To safely consider those plumes which are definitively near the surface (i.e. never above the boundary layer) a plume height below 200m would roughly corresponds to the boundary layer maximum in the middle of the day (Guo et al., 2019). However, due to the measurement uncertainty of the MISR heights being between 250m and 500m, instead the percentage of total plumes with a height below 500m is chosen, which is found to have a total percentage of respective plume heights of: global (11%) and a range from a minimum of 0.68% in Southern Africa to 49% in Argentina. Given the diversity of these results, there is a need to more deeply

understand the driving factors across all of these different regions, as well as the importance of biomass burning in terms of transporting aerosols through the boundary layer.

Second, we perform a comparison across the different daily time series of measured aerosol heights, CO column, and NO$_2$ column as aggregated from January 1, 2008 to June 30, 2011 over all of the biomass burning regions Fig. S1. We consider the burning season to be when we observe aerosol plumes and a peak in at least one of the CO and/or NO$_2$ column measurements. Furthermore, MISR has a relatively narrow swath, not providing daily coverage to all points, coupled with a morning overpass time which may lead to negative bias in some regions and positive bias in other regions in terms of observed fires. This combination allows us to clearly demonstrate that the observed smoke peaks are in fact due to burning of a significant amount of material, and are true cases of biomass burning, while not possibly being fully representative of all biomass burning events. In the observed cases, the peak occurs from November to March in Central Africa, Midwest Africa; June to September in Central Canada, Eastern Europe, South America; April to July in Central Siberia; May to December in Southern Africa, Northern Australia; January to April in Northern Southeast Asia; March to September in Siberia and North China; April to September in West Siberia. In addition, the length of the peak burning time is also an important consideration which varies greatly across the different regions. The length of the total number of burning days from the three and a half years of data is an average is 108 days, with a minimum of 14 days in Eastern Siberia and a maximum of 388 days in Southern Africa.

Next, we look at the impact of FRP measurements and buoyancy in terms of the plume height distribution. In general, the higher the FRP, the higher the plumes should rise. However, these measurements seem to include a larger number of total measurements into the lower free troposphere than previous plume rise model studies have been able to account for. From our measurements, we notice that the FRP (as computed on average over 1.1kmx1.1km grids where a fire exists) has a global mean of 37.7W/m$^2$ and a reginal minimum and maximum of 31.1W/m$^2$ (Siberia and North China) and 82.6W/m$^2$ (Central Canada) during the respective biomass burning seasons. Analyzing the extremes of the FRP, leads to a top 5% of measured FRP of 132W/m$^2$ and a bottom 5% of measured FRP of 8.5W/m$^2$.

Based on previous work, we would expect a general plume rise model to not be able to match the observed heights well under these conditions, since the observed FRP is too low (Cohen et al., 2018; Gonzalez-Alonso et al., 2019). The high end of the FRP range of the observations in this work is not considered to be very hot in terms of fires, which should in theory help to reduce the known plume rise model bias of underpredicting very strong FRP conditions, leading to an overall improvement in the plume rise model as analyzed in this work, as compared to when it is less constrained. As expected, there are more plumes found above the boundary layer in the measurements corresponding to very high FRP values than in the case of very low FRP values, although more importantly, there still are plumes found over the boundary layer in both cases, which is not expected based solely on the plume rise model formulation. One possible explanation for this phenomenon is that the biomass burning occurring during the times of year where there is a negligible impact on the atmospheric loadings of NO$_2$ and/or CO

is significantly more energetic and therefore has a very different height profile, as compared to the times when the most emissions of $NO_2$ and/or CO are produced. Another explanation is that there is additional forcing which are also playing a role in terms of the aerosol plume height rise that are independent of the FRP. Yet another possibility (Mims et al., 2010; Val Martin et al., 2018) is related to there being some type of problem with the presentation of the nature of the land-surface itself, since fires occurring in heavily forested and agricultural areas are likely to have significantly different vertical distributions. On top of this, there may be partially filled pixels in the remotely sensed measurements (Kahn et al., 2007; Val Martin et al. 2012). Finally, it is possible that the intense aerosol loadings themselves are leading to absorption of a significant amount of the IR radiation which is in turn biasing the FRP measurements too low (Cohen et al., 2017; Cohen et al., 2018).

It is also possible that there are significant differences to be found in the non-linearity between FRP and the wind speed. Interestingly, if the horizontal wind speed is quite high when the air passes over the fire source, it will cause turbulence and vortices, resulting in a lifting force. On the other hand, if the wind speed is too high, it will bend the plume's momentum and reduce the upward transference based on any initial vertical injection velocity. Furthermore, the wind speed may have different relationships with convection, which itself plays a dominant role in the rise of the plume. Given these effects, we do not directly consider wind-speed and the plume rise height independently, only within the confines of the plume rise model.

Since there are many underlying direct and indirect theoretical physical and chemical connections between the loadings of the CO and $NO_2$ and the overall plume heights from MISR, we want to investigate this possibility more deeply. To make this comparison, we first looked at the entire time series, not only those periods during which the measured aerosol heights obviously had an impact on the atmospheric loadings of the CO and $NO_2$. Next, we selected days which had data from all three measurement sources: MISR, MOPITT, and OMI. Furthermore, since we could not find such a paper in the literature, we have decided to keep the relationship open and simple, without worry of over-constraining any relationship found. In theory, the injection height of the aerosol plume is related to the emission of smoke in the wildfires, since this is a function of the amount of heat released. Therefore, we would expect that higher emissions of CO and $NO_2$ should correspond to higher heights of aerosols. However, the formation mechanisms of these two trace gasses is different, with CO a function of oxygen availability (and possibly surface wetness), while $NO_2$ is a function of the temperature of the burning. Furthermore, very high co-emitted levels of aerosols with the very high levels of trace gasses could also lead to a change in the vertical profile of the heating (Freitas et al., 2007).

To ensure that the variables are relatively independent, our analysis only considers only three mixtures of these species: the independent concentrations of CO, $NO_2$, and the ratio of $NO_2/CO$. We then investigate how changes in the loadings of $NO_2$, CO, and the ratio of $NO_2/CO$ are associated with changes in the height of the plume. Furthermore, we need to consider the more extreme conditions in addition to the means, and are particularly interested in seeing how well loadings of the CO and $NO_2$ can be used to model those conditions where the plume heights are extreme.

In all of the regions of the world, with the exception of the case of $NO_2$ over Siberia and Northern China, we have a case where the mean value of the CO and $NO_2$ measurements is higher over the set of points where the actual FRP measurements were made, than over the region as a whole Table 1. This is the point of this work, since we want to focus on the measured values from MOPITT and OMI which correspond to the same spatial locations as the measured FRP. This makes sense, since the magnitude of emissions from fires is very large compared with the non-burning season and/or surrounding areas. However, the differences in the CO are in generally smaller than for $NO_2$, which is further consistent with the fact that the lifetime of CO is much longer than that of $NO_2$. Thankfully the case is well understood over Siberia and North China is because there are some known significant urban areas nearby to the burning regions. Furthermore, this exception occurs in winter, where we know there is a significant enhancement of $NO_2$ emissions due to the increase in urban biomass burning to offset the brutally cold winter conditions.

Over these fire-constrained points, we find that the variability of both CO and $NO_2$ remains very low when computed on a point-by-point basis. On the other hand, over the entire regions, the variability of the point-by-point measurements of both $NO_2$ and CO are much higher. This is in large part due to the rapid changes in different land-use types in different parts of the regions of interest being studied (consistent with Cohen et al., 2018). These results are based on the statistics of more than 67000 daily MISR measurements. Therefore, for the remainder of the work, we only use the data for the $NO_2$ and CO which are obtained at points where FRP measurements exist.

Note that the measurements and the results here are looking at the aerosol heights measured over small spatial and temporal domains. We are looking to analyze the impact of the initial plume rise, and any very rapid adjustments in the atmosphere. The plume heights, both measured and modeled are not consistent with large-scale transport due to meteorology, factors enhancing the stability of a layer or changing the chemistry within a plume. They certainly are not receptive to a Lagrangian type of modeling effort, which is supposed to be focused on the air itself and in particular air at the large scale. Therefore, the results given here show here are the best methods currently used to reproduce the spatial distribution of aerosol plumes produced by wildfires.

**3.2 Plume Rise Model Applied to MISR and Meteorological Measurements**

The annual average global total cumulative FRP from 2008 to 2011 is 209 MW, based on more than 16000 measured MODIS fire hotspots. Overall, the measured FRP has been shown to be on the rise in recent years (Cohen 2018; Freeborn et al., 2014), although there is still a fundamental and significant amount of underestimation based on the current measurement techniques (Giglio et al., 2006). The plume rise model in theory should take these FRP values, and combine them with knowledge of the vertical thermal stability and the wind speed, to approximate the height to which the plume ultimately rises at equilibrium with its environment.

However, in reality, direct and semi-direct effects are not considered when using the simple plume rise model, although they are known to be important (Tao et al., 2012). Therefore, a different approach which attempts to take these forcings and/or the underlying aerosol loadings into account may lead to a better representation of the plume height rise, if such a model can be parameterized. Furthermore, the plume rise model relies on the atmospheric stability, and therefore does not take into account rainfall, changes in fire burning, in-situ chemical and physical production and removal, as well as the afore mentioned interactions radiatively between the aerosol and the atmospheric environment. This finding is consistent with evidence that the vertical plume rise and distribution of tropical convective clouds is sometimes dominated by in-situ heating and turbulence even more so than the initial heat of condensation (Gunturu, 2010).

All of these shortcomings aside, the use of simple plume models is the current scientific standard approach, and therefore we will apply it here as well. This is done by first aggregating the daily statistics of the vertical aerosol height over all parts of each region of interest Table 2. Direct comparisons are made between the modelled heights and the measured heights, and we find that 5 of the 14 regions studies in this work were shown to have a good match: West Siberia; Alaska; Central Canada; Argentina; Eastern Europe, where the modelled (and measured average heights) respective are 0.79 km (0.95 km) 1.39 km (3.03 km), 1.73 km (2.19 km), 0.65 km (0.25 km), 1.27 km (2.67 km).

Next, we look at the difference from day-to-day at each of the sites which has a mean value less than or equal to 0.25 km. Using these results, we find that the mean daily difference between the plume rise model and the MISR measurements as a whole show a large amount of variation, with a global average of 0.44 km, a maximum of 1.13 km (in West Siberia), and a minimum of 0.04 km (in Argentina). Across all of the different regions we find that the plume rise model underestimates the plume height. Furthermore, we find that the differences between the plume rise model and MISR are not normally distributed, with higher values not being able to be reproduced under any conditions, strongly indicative of a bias, in that somehow the largest, hottest, or most radiatively active fires are those being not reproduced well by the plume rise model. In addition to this, we compute the RMS error (Table 3) as a way of quantifying overall how well the model and MISR match. The RMS is found to be considerably larger than the difference of the means, indicating that a small number of extreme values are dominating the overall results, which were found to be 0.67 km, 0.88 km, 1.36 km, 0.40 km, and 0.85 km in the respective five areas.

To more carefully determine the extent of any bias, we analyze the PDF of the model and measurement results Fig. S2. This approach yields a clear determination that the plume rise model consistently underestimates the measured injection height, with the underestimate ranging from 6% (in Argentina) to 66% (in Southern Africa), and a global average of 33%. However, if we constrain ourselves to those fires occurring only in heavily forested regions, the average underestimate is reduced considerably to 11%. On the other hand, if we look across Africa as a whole, we find that the underestimate is on average 52%, a finding which deviates more from the measured aerosol vertical distribution than previous global studies (Val Martin et al., 2018) as well as those over Southeast Asia (which previously has been considered one of the world's worst performing regions for such plume rise models, such as Reid et al., 2013; Cohen, 2018). The only region over which the finding may not be

statistically relevant is in Alaska, where the difference between regression measurements are all constrained to within a 500m height band, which falls into the MISR measurement uncertainty measurement range.

Furthermore, even though the plume rise model leads to a low bias compared with the measured height, it is still not ideal for very low plumes which are found near the surface. The plume rise model tends to instead uniformly overestimate the amount of aerosol found in the upper parts of the boundary layer from 0.5km to 1.5km, while at the same time not providing any reliable amount of prediction for the cases where there is a considerable amount of aerosol under 0.5km. For example, the plume rise model is sometimes a good fit for aerosol heights under 0.5 km such as in West Siberia and Eastern Europe (where 23.5% and 12.3% of the measurements are under 0.5km and 27% and 13.6% of the plume rise model heights are under 0.5km, respectively). However, in other locations, the plume rise model grossly overestimates the amount under 0.5km such as in Central Africa and East Siberia (where 3.6% and 17.9% of the measurements are under 0.5km and 20.5% and 51.0% of the plume rise model heights are under 0.5km, respectively). In the case of Argentina there is a slight underestimate of the 0.5km heights by the plume rise model (49.4% of measurements and 30.1% of the plume rise model heights). One of the reasons for this is that in general the plume rise model tends to underestimate the results from 1.5km to 2.5km, and cannot reproduce results reliably at all above 2.5km. This is partly due to the effect that the FRP values are too low, and possibly due to other processes occurring in-situ which further lead to buoyancy and/or convection.

A few special regions of interest have been observed when comparing the plume rise results with the measurements. In Southern Africa plumes cover 9763-pixels or 19% of the total research area, and therefore are extremely representative of the overall atmospheric conditions. What is observed is that there is almost no aerosol (only 5.9%) present close to the ground (from 0km to 1km). The vast majority of the aerosols, 92.6%, are concentrated from 1km to 3km. Furthermore, we observe that the time series of both CO and $NO_2$ loading is significantly higher than other regions Fig. S1. This finding is completely the opposite form the plume rise model result, which shows that most of the pollutants (97%) are concentrated in the range of 0-1km, while almost none (3%) is found from 1km to 3km. There are a few reasons for this finding. First of all, when both CO and $NO_2$ loadings are high, the aerosol concentration and AOD will also be high, since they are co-emitted at roughly similar ratios from the fires. This in turn will both lead to a further underestimation of the FRP due to the outwelling infrared which is partially absorbed by the aerosols, as well as provide a further uplifting energy source due to the semi-direct effect (Tao et al., 2012; Guo et al., 2019). In other words, the assumptions underlying the plume rise model may not be completely relevant or dominant over this region under these conditions.

A second special region, which completely contrasts with Southern Africa is found in Argentina. In this region, a much smaller amount of the total research area is covered in plumes of 1063-pixels or 2.1%. A large amount of the total aerosol (83.8%) exists below 1km, while only a small amount (5.1%) is found above 2km. In this case, the plume rise model achieved its best match globally, with a large amount (92.2%) found below 1km and a small amount (0.35%) found above 2km. Furthermore, the loadings of CO and $NO_2$ are both considerably low as compared to other regions studied in this work. It is

under these relatively lesser polluted conditions, where the fires are fewer and/or less intense, where a lower amount of total material is being burned on a per day basis of time over the total surface area burning, or where the meteorology and the vertical thermodynamic structure of the atmosphere are more uniform, that the plume rise model can achieve its best results (Table 4, Fig 6 and Fig S6), and thus that the plume rise model is reasonable to use in such a region. Although it is still obvious that even in this best result case, that the plume rise model is fundamentally biased towards the aerosol vertical distribution being too low, especially the amount into the free troposphere.

As we have observed, the simple plume rise models based on Briggs, 1965, are useful under specific circumstances. This is especially the case when the atmosphere is relatively stable, the total loading of pollutants is not too large (i.e. there is fewer fire masking and less of the semi-direct effect to contend with), and where the density of fires is lower (and hence there is less overall buoyancy changing the atmosphere's dynamics). On top of this, more flat and uniform areas are less likely to have local convection, further leading to an improvement of the effectiveness of the simple plume rise model. It is for these many reasons why we find that the simple plume rise model does not provide an ideal fit over many regions, and for this reason, we propose a simple statistical model as an alternative.

**3.3 Regression Model Applied to MISR, OMI, MOPITT and Meteorological Measurements**

Since plume rise models rely solely on information related to fire intensity and meteorological conditions in order to compute an aerosol injection height, we want to build a relationship that also includes the net effects of pollutants as well. Therefore, we introduce and globally apply seven different combinations of relationships between FRP, Wind, CO, $NO_2$ and injection height Eq.1-7. Different combinations of CO and $NO_2$ are applied to the linear regression model. CO and $NO_2$ are independently mixed with the meteorological terms in Eq.4 and Eq.5, while they are jointly mixed together with the meteorological terms in Eq.1. A non-linear weighted variable of $NO_2/CO$ is mixed on its own with the meteorological variables in Eq.6, while it is mixed with either one of CO and $NO_2$ in Eq.2 and Eq.3. The reason for this is that there is a significant physical relevance for determining how much $NO_2$ is emitted per unit of CO, which is a strong function of the fire temperature as well as Oxygen availability. This set of models is capable of providing a clear relationship between the response of either or both of CO and $NO_2$. Such an approach allows for us to examine the strengths and weaknesses of each combination in terms of the spatial-temporal distribution of the measured heights, as well as the contribution to the absolute magnitude.

The regression model solely containing $NO_2$ is an approximation of the concept that the heat of the biomass burning should have an important role to play in terms of the plume height. Furthermore, using $NO_2$ in this way helps to get around the inherent underestimation of FRP. The regression model solely containing the CO is a proxy for the concept that the mass of biomass burned should make an important contribution towards the plume height. Inclusion of the CO term is also a way to get around the underapproximation of the total burned area, or of any significant contribution from underground burning.

The average statistical error and average statistical correlation (coefficient of determination, $R^2$) between the datasets used to determine the best-fit coefficients for α, β, γ, δ and ε are displayed in Table 2. While a comparison of the time series of the region-averaged injection height was made over all 14 regions, only those regions passing a level of quality control as described below are retained. First, different linear combinations are evaluated for their correlation against the MISR measurements, with an optimal combination selected and considered to be a success only if $R^2>0.2$ and the $P<0.05$. Furthermore, we compare the modeled average injection height in an absolute sense to the measured values, and retain the data if the difference is smaller than 0.25km. Based on these results, the best-fit model-predicted injection height and the measured averaged injection height were found to be reasonable only at 8 different sites.

In general, when CO or $NO_2$ or both are included in these different combinations for these regions, the normalized coefficients of CO and $NO_2$ have a larger value than the respective normalized coefficients of FRP or wind speed. This means that when these variables occur simultaneously, the contaminants have a stronger influence on the final injection height of the plume. This is found to especially be the case in regions which have higher loadings of pollution. The regression model with the non-linear combination of the two is a proxy for the argument that it is the ratio of the heat to the total biomass burned that is an essential physical consideration to take into effect. Furthermore, this final case provides some weight to the concept that a small change in the vertical column concentration may have a stronger than linear effect, as is evidenced by (Ichoku et al., 2008; Zhu et al., 2018), such as in terms of absorbing aerosols in the vertical column altering the ultimate vertical distribution.

This comparison also is found to be valid in regions which in general are less polluted. For example, even in relatively clean Central Canada, the linear combination of $NO_2$ and the ratio of $NO_2$ and CO produces the best fit, with the coefficient of $NO_2$ being roughly an order of magnitude larger (at $3.2x10^3$) as compared to the coefficients of FRP and Wind (which are respectively a magnitude of order smaller, at $3.3x10^2$ and $-2.3x10^2$).

Due to the fact that $NO_2$ and CO have very different lifetimes in the atmosphere, a fire-based source is expected to have a high level of both CO and $NO_2$ close to its source, which decays as one heads away in space from the source. This decay should be a function of the wind direction as well, as both the CO and $NO_2$ upwind will not have a significant source, but downwind the CO will have a significant source, as shown in Fig. 5. We find that our results are consistent with this theory. First, we have found that the regions that have the highest $NO_2$ at the same time as the MISR measurements are made, also have a very strong overlap well with the locations of the MISR plume heights. We further determine this to be true for every year on a year-by-year basis (Fig S1). Second, we find that the higher values of CO match well with the year-to-year locations of MISR fires (or downwind thereof) at most of the sites, including in Alaska, Central Canada, Central Siberia, East Europe, East Siberia, Northern Southeast Asia, Siberia and North China, and South America. As expected, there greater smearing away from the source regions. As expected, this is due to the fact that the lifetime of CO is much greater.

Following these ideas, the idea of characterizing the by the ratio of $NO_2/CO$ is found to nicely separate the data into three different groups, based on the bands generated by the central 80% of each respective region's $NO_2/CO$ pdf. Group 1 consisting

of Siberia and North China & Central Canada, has a $NO_2/CO$ range from $1 \times 10^{-4}$ to $9 \times 10^{-4}$. Group 2 consisting of the remaining regions, has a $NO_2/CO$ range from $2 \times 10^{-4}$ to $15 \times 10^{-4}$ to $20 \times 10^{-4}$. Group 3 consisting of South America, has a $NO_2/CO$ range from $6 \times 10^{-4}$ to $43 \times 10^{-4}$. This strong differentiation is consistent with the ratio of $NO_2/CO$ representing a physical meaning, but being a single, continuous variable connected with the temperature of the burning, the wetness of the burning material, the latent heat flux, and the type and amount of biomass being burned.

Furthermore, in terms of changes in time, a climatology of CO should be slightly higher due to the added emissions from the fires, but the $NO_2$ should be much larger than the climatology, since there is little to no retention in the air, as demonstrated in Table 1. To account for this, we have also looked at the difference between the fire times and the long-term climatology. Over regions which are urban and hence contributing randomly to the variance, we expect the differences to be smaller than due to the fires, and this is observed clearly as well. On top of this, the $NO_2$ column loading and the ratio of the $NO_2/CO$ column measurements over only the selected grids which have available of FRP measurements, and over the larger regions as given in Fig. 5 are found to generally be consistent, with the ratio found to be more so (Table S2). This indicates the $NO_2/CO$ column ratio over the fire regions tends to be consistent with the fire plumes as a whole, and is not found to be significantly influenced by urban sources of $NO_2$, which would lead to a vastly faster chemical titration of $NO_2$ compared to CO. All of these results are also shown to be consistent with recent work (Cohen, 2014; Lin et al., 2014; Lin et al., 2020a), showing that the characteristics of the spatial-temporal variability of fires is quite different from that of urban areas, and has a much higher variability both week-to-week and inter-annually.

**3.4 Comparison between the Plume Rise Model and the Regression Model**

The results in Table 3 indicate that inclusion of either one of CO and $NO_2$ or in some cases both, always provides a better fit to the measured vertical heights when using the regression aerosol height rise model, as compared with those model cases where the loadings of the gasses are excluded. In addition, the fit is better over a larger number of regions (8 regions versus 5 regions), details are shown in Fig. S3. What we observe is that the regression model does relatively better in regions which are more polluted, while the plume rise model does relatively better only in regions which have very low amounts of burning in terms of FRP. A detailed look at the day by day values from the MISR measurements of aerosol height, the regression model of aerosol height, and the plume rise model are given in Fig. 3.

As shown in Fig. 3 there are three regions where both modeling approaches work well. In West Siberia, the regression model shows more stability than plume rise model, with the results more narrowly concentrated around 1000m. Furthermore, the results are mostly found within the range of the measured variation. The plume rise models results are also relatively stable, although more dispersed in general than the regression model. Overall the RMS is 0.47 between the measured values and the regression model, while it is 0.67km between the measured values and the plume rise model. A similar set of results is found in Alaska, with the RMS for the regression modeling being 0.88km and that of the plume rise model being 0.77km. The major

difference here is that the plume rise model results have a variance higher than that of the measurements (SD of 0.91km for the regression model and 3.03km for the plume rise model). In the case of Central Canada, although both modeling approaches have a decent fit, there is a clear difference between their overall performance. In general, the results of the plume rise model (1.73km) are biased significantly lower than both the measurements (1.97km) and the regression model (2.13km), while there is little bias between the measurements and the regression model. To make this point clear, only roughly 7.9% of the measured results are outside of 1 standard deviation from the measured mean, while 50% of the plume rise model results and 43% of the regression model results are found to be outside of the 1 standard deviation from the measured mean. Note that this is the sight which has the highest RMS error and still yields a successful fit for both modeling approaches. Details are given in Fig. S4.

In some of the more highly polluted regions, the regression model showed a decent performance, while the plume rise model did not. The overall goodness of the fit of the regression model is reasonable in the cases of South America, Siberia and North China, Northern Southeast Asia, and Northern Australia. This is because these areas emit large amounts of CO and $NO_2$, in some cases solely during the biomass burning season, and in other cases due to a combination of biomass burning and urban sources. Overall in these more polluted regions, the regression model is found to have little bias (respectively -0.02km, -0.20km, -0.22km, and 0.15km), which helps to establish the predictive ability of using the gas loadings in terms of predicting the vertical distribution of the aerosol heights.

Although the vertical distribution of aerosol cannot be successfully simulated at all sites by using the regression model approach, at the sites where it provides a reasonable fit, it seems to do better than the plume rise model approach. This is further found to be true in the case where the data at the high end of the $NO_2/CO$ ratio profile are considered. This improvement is found in terms of both the bias and the RMS under all conditions, and even more so at the respective top and bottom 10% of each respective range of $NO_2/CO$ ratio, in which the subset of regression model heights performs much better than the respective plume rise model heights when compared with the MISR height distribution (Figure S7).

These findings are consistent with real true world conditions, where there is a significant impact of co-emitted aerosols and/or heat, and these results with the $NO_2/CO$ ratio would hint that higher burning temperature conditions, or fewer oxygen limited conditions, may be important driving forces. These changes either directly alter the heating throughout the profile as well as indirectly introduce a negative bias on measurements of the FRP below. No matter the underlying specific reasons, overall, we find that the regression model approach yields at least as good if not a more precise representation of the plume rise height as compared with the simple plume rise model. However, combining the two approaches yields the best overall result, since there are some locations in which each approach is better than the other approach.

What is most important to note is that in some of the regions, none of these simple approaches work. This is particularly so for when the measured distribution of the aerosol heights is when there is a diverse set of sources. For example, in Africa there are significant sources from biomass burning, as well as from rapid urbanization and burning over many different land use types and under many different types of conditions. Another potential problem occurs when there may be a significant

amount of smoke which has been transported from another region, such as the exchange of smoke between the Maritime Continent and Northern Australia. Furthermore, both approaches will not tend to work well under conditions in which the atmosphere is not highly stable, or has a high variation in weather conditions. Under these conditions, a more complex modeling approach and the improvement of measured fire data.

**3.5 Comparison between MISR and the three Models**

A comparison between the overall performance of the plume rise model, the regression model, and MERRA leads to a few conclusions (Table 3). First of all, where the regression model exists, it reproduces the MISR height better than both the plume rise model and MERRA. This includes over regions where the overall RMS error is very low such as Eastern Siberia and South America, as well as regions where the overall RMS error is large, such as Central Canada. This is true including over regions in the Arctic as well as in the tropics. Secondly, over the regions in which the regression model does not exist, MERRA provides a better reproduction of the MISR height than the plume rise model in all cases, except for over Argentina. Perhaps this is true because of the fact that although MERRA uses data assimilation and a plume rise model type of code built in, the sharp height rise of the Andes Mountains and high cloud cover over this region lead to challenges that the global MERRA model cannot handle well. The second possible explanation is that the overall height of the plume is very low over Argentina and the local meteorology and FRP values are quite similar, which play to the plume rise model's strengths.

Furthermore, comparing the performance of the plume rise model, the regression model, and MERRA at different percentiles of height leads to additional conclusions (Table 4). On one hand, the regression model is the only one which does not have an obvious bias versus MISR measurements, with the regression model sometimes overapproximating and other times underapproximating different geographic locations at different height levels. In fact, the results at the median and 70% height levels are an excellent fit for 4 of the 8 different regions. On the other hand, both the plume rise model and MERRA have obvious biases. The plume rise model is almost always too low, with the only exception being its ability to model 6 of the 14 regions reasonably well at the 10% height level (i.e. the bottom of the plume). However, in the case where the 10% level is higher than other cases, such as a very narrow distribution, the plume rise model still dos a poor job. MERRA is almost always too high, with it performing best at only South Africa and East Europe. Furthermore, the results from the plume rise model tend to also be narrower than the data, while the results from MERRA tend to be broader than the data. The results of MERRA being broad, as demonstrated clearly in Fig. 4, are not due to a high inter-annual variability, which actually barely exists in the MERRA dataset as compared with the regression model and MISR, but instead due to too much aerosol being found too high in the atmosphere, as well as too much aerosol being found at the surface.

The MISR data, regardless of the region, shows some amount of inter-annual variability. This ranges from a minimum over East Siberia and Siberia and North China, to a maximum over Central Canada and Northern Southeast Asia. On the other hand, MERRA shows only a very small variation anywhere, with most of the years exactly the same as each other. The amount

at the surface is always much larger than found in MISR and the amount in the middle free troposphere is also much larger than in MISR. The largest variation in MERRA is found in Central Canada, Alaska, and Northern Australia. All of these are regions which are relatively cloud free and have vast amounts of ground stations, and therefore will have a large amount of the total MERRA model contribution from reanalysis data.

In the case of East Siberia there is only burning observed by MISR in 2 of the 4 years studied here, although these two different years have quite a different distribution. In 2008, the aerosol is limited in height to under 1000m, while in 2010, the aerosol has a peak height at 1000m and a significant fraction up to 2000m. In the case of Siberia and North China, the peak ranges from 800m to 1200m and the maximum ranges from 2200m to 3000m. MERRA shows no burning at all in East Siberia, with a completely flat profile all 4 years, and a consistent burning year to year, with the aerosol all confined to 1000m and below over Siberia and North China. In terms of the regression model, the fact that there is a good fit is supported by Fig. 5. As can be observed, all of the fire data points occur in regions of high CO and the vast majority also occur in regions of high $NO_2$. In Siberia and Northern China, the findings in both of the years in Fig. 5 lend support, albeit from two different perspectives. The first is that the fires always overlap with regions of high CO, and that in the 2011, one of the major differences is that the region in the middle has low CO and no fires, which were both present and highly polluted in 2008. The $NO_2$ is always high over the southern region, and is never very high in the central or northern regions, likely due to the intense cold air present in these regions altering the $NO_2$ chemistry.

Over Central Canada the MISR data shows peaks or sub-peaks at 1000m in 2008, 2800m and 3200m in 2009, 2000m in 2010, and 1000m and 2600m in 2011. In many of these years the amount located in the free troposphere is much larger than the amount in the boundary layer. Yet, even though this is the region in which MERRA has the most inter-annual variability, in all cases, the vast majority of the aerosol is found below 1000m. Furthermore, no peaks or subpeaks are found anywhere above the surface. Finally, MERRA only shows 1 year to be considerably different from the others, whereas the MISR data shows that all 4 years are quite different. By looking at Fig. 3, we can see that the regression model on some days underestimates the plume, on some days overestimates the plume, and on some days is nearly perfect. There is no bias, and the fact that it is able to capture the range of values over all 4 years indicates that the performance is not only better on average, but as well at capturing the inter-annual variation over this region. This finding is further supported by Fig. 5, where all of the MISR fire points in Central Canada in 2010 are found in high CO pixels, and most of the MISR fire points are also found in high $NO_2$ pixels. This demonstrates that the vast majority of the MISR plumes are local in nature and actively connected with the ground (due to the short lifetime of $NO_2$), are in relatively cloud-free regions where these remotely sensed platforms will work, but not necessarily MODIS which may be blocked by the high AOD levels, while also being in regions which are clearly heavily polluted by CO during these times, but are not normally so.

The MISR measurements over Northern Southeast Asia show the majority under 1000m but a second peak around 2500m in 2008, the peak at 2500m and a large amount up to 3200m in 2009, the peak was spread from 500m to 2500m in 2010, and

peaks at 1000m, 1200m, and 2200m in 2011. This huge amount of inter-annual variability is not at all captured by MERRA, which is consistent with other recent findings over this area of the world demonstrating that many products based on MODIS tend to have problems (i.e. Cohen 2014, Cohen et al., 2018). However, the regression model performs well over this region as over all of the years, with measurements again showing an unbiased representation in all 4 years of the height, with some days high, other days low, and some days nearly perfect. This is in part demonstrated clearly in Fig. 3 and Fig. 5 by the fact that the MISR fire points occur over the highest loadings of CO and $NO_2$ found among any region, anywhere else in the world, as observed in this study.

In terms of the magnitudes of the vertical temperature gradient (dT/dz) and the vertical wind speed at the surface, we have not found any correlation or relationship between the cases in which the regression model performs better or worse. Even considering those cases in which there are extremely atypical values in these variables, such as positive temperature gradients (i.e. an unstable atmosphere), or negative temperature gradients which are more negative than the -9.8 K/km rate which is the pure dry air thermodynamic limit (i.e. extreme stabilization due to intense aerosol/cloud cooling), as observed in Fig. 6. This provides a further piece of support to the idea that the regression model works well under conditions where there is some local non-linear forcing in the system which is not being taken into account, whether it is a coupled chemical, aerosol dynamical/size, radiative-dynamic, thermodynamic, or direct/semi-direct/indirect type of aerosol effect, all of which are being accounted for to some degree by the loadings of $NO_2$ and CO, but which are missed by the model underlying the meteorological reanalysis data (e.g. Cohen et al., 2011; Wang et al., 2009).

However, it does seem that under the conditions where the regression model was not able to be formed, that there are some important differences in terms specifically of the vertical temperature gradient variable. In specific, in the cases in which the value of dT/dz is either more negative than -9 K/km or positive, that the MERRA results are far better than those from the plume rise model, as compared to not under those conditions. However, such cases only account for 15% or fewer of the total cases observed in this study, and therefore do not play an outsized role.

**4 Conclusions**

This work quantifies the measured values of the aerosol vertical distribution over biomass burning areas of the Earth on a daily basis from January 2008 through June 2011. We find that there is a significant amount of total aerosol which reaches the free troposphere, as well as large amounts which are not uniformly distributed throughout the boundary layer, both of which are not readily explained by first order theoretical approximations and present-day community-standard models.

To address these issues, we introduce a new approach, based on remotely sensed measurements of fire properties, wind, and column loadings of $NO_2$ from OMI and CO from MOPITT to constraining the aerosol heights over different geographic regions. This approach is based on the physical concept that the emissions of aerosols and the height to which they rise should

be related to other co-emitted species like $NO_2$ and CO and the co-emitted heat, which is also a function of the ratios of $NO_2$ and CO produced by the burning. Our results are compared against both the measured MISR height values as well as basic plume rise model computations using the same fire radiative power and meteorological datasets.

Our results indicate that our new method reproduced the measured values significantly better over much of the world in terms of reproducing the measured vertical distribution as compared with the simpler plume rise approach. In specific, we find that applying the plume rise model leads to a model underestimation of the measured MISR heights overall, whereas our approach, where it works, does not exhibit such a bias. This finding is consistent with the fact that FRP is underestimated globally, in part due to clouds and aerosols, and in part due to sampling and other issues. We also find that the plume rise model tends to be too narrowly confined compared with the regression model and the modeled results. However, the plume rise model does better in terms of reproducing the aerosol injection height when it is solely contained and well mixed within the atmospheric boundary layer, but for higher altitudes, the model capability is poor. The average underestimation of the plume rise model injection height is 33%. On the other hand, the regression model has an overall improved the accuracy of the measured results, in particularly doing a better job reproducing results in the free troposphere. The regression model is also more widely applicable around the globe, with the number or regions successfully simulated increasing from five to eight. As we have demonstrated, the impact of $NO_2$ (as a proxy for the burning temperature) is always essential, and the impact of CO (as a proxy for the total biomass burned) is usually essential as well. We further have shown that the simplest regression model, the approximation of the plume rise model, never yields the best fit to the data.

In specific, we find that the plume rise model works well in regions which are not frequently cloud covered during the local biomass burning seasons, in particularly so over non-tropical forested regions. In specific, the plume rise model has its greatest successes in Alaska (RMS error of 0.77km for the regression approach versus 0.88km for the plume rise model approach), Argentina (the regression model approach does not succeed versus and RMS error of 0.40km for the plume rise model approach), and Western Siberia (RMS error of 0.47km for the regression approach versus 0.67km for the plume rise model approach). In most of the other parts of the world, the regression model approach is much better at reproducing the vertical distribution than the plume rise model, even including some major extreme events including the release of aerosols into the stratosphere, and tends to do so with a reasonably lower RMS error and low standard deviation.

One of the major advantages of the regression model approach is that it is more capable of picking up those cases where aerosols are lofted into the lower free troposphere, and another advantage stems from its ability to reproduce better those cases where the near surface is clean, but the upper part of the boundary layer is polluted. In the cases of Eastern Siberia and Amazon South America, we find that the regression model performs reasonably well, while the plume rise model does not succeed. In the case of Northern Australia, the regression model is capable of reproducing the aerosol height with a relatively reasonable set of statistics, although the measurements in this region are found to be very unique; sometimes the plume is mainly concentrated in the lower free troposphere and is local in nature, while other times it is found in the upper troposphere and

lower stratosphere, in which case it is thought to be transported from the Maritime Continent. We also find that the regression model works well in two other special cases. The first is the case of Siberia and Northern China, where there is a considerably large amount of local urban pollution which is mixed into the biomass burning plumes. The second is the case of Northern Southeast Asia, where there are both large amounts of local pollution as well as considerable issues with extensive cloud cover.

Our results show clearly that where we can successfully form a regression model, that it performs better than both the plume rise model and MERRA. The specific forms of the regression model that are the best are those which have $NO_2$ or a combination of $NO_2$ and CO (in particular when the non-linear term $NO_2/CO$ is considered). These results are consistent with our hypothesis and literature review that show new forms of non-linearity relating plume rise height to factors influencing buoyancy, radiative transfer, and energy transfer in-situ, and/or biases in remotely sensed measurements of FRP and land-surface products are important. Such are not considered in the present generation of plume rise models (including the global-scale models underlying MERRA). In the cases where we cannot form a regression model, we find that MERRA performs better than the plume rise model everywhere, except for Argentina, which has a unique high mountain just upwind in the Andes, coupled with a very low overall height, all of which are disadvantages for the models underlying MERRA. In general, this shows that improved model complexity and data assimilation doe produce a better result, as expected.

We propose the results as a first step of a new approach to parameterization that my help us to move forward in terms of improving our ability to reproduce heights of fire plumes for regional and global scale modeling and analysis studies over many different periods of time. We believe that our sample dataset is currently not sufficiently long to form an ideal fit, and hence thought that excluding data to self-compare was not an ideal use of the very limited resources we had. We do hope that as more new datasets are released, the community will have access to more relevant input data, and as more MISR plume height data is released, the community will have more access to better understand the vertical distribution of height.

Based on these results, including over those regions where none of the results yield a satisfactory response, we have come up with a list of recommendations for how to improve the reproduction of the vertical aerosol distribution in the future. First, improving the accuracy of FRP measurements, especially so under cloud and heavily polluted conditions. Secondly, improving the ability of simple models to compensate for the impact of local-scale radiative forcings, deep convection, aerosol-radiation interactions and aerosol-cloud interactions. Thirdly, the finding that the $NO_2/CO$ ratio is extremely important in terms of matching the vertical distribution works to address the larger community issue of flaming versus smoldering in a more smooth and precise way, opening the possibility of a new continuum approach to consider burning wetness, temperature, and heat. Based on our overall results, we believe that an improvement can be made to the current generation of GCMs, atmospheric chemical transport models, and remote sensing inversions, all of which depend on a more precise knowledge of the aerosol vertical distribution.

**Author Contributions**

Shuo Wang: data curation, formal analysis, investigation, software, visualization, writing – original draft.

Jason Blake Cohen: conceptualization, funding acquisition, investigation, methodology, project administration, resources, supervision, validation, writing – original draft, review & editing.

Chuyong Lin: data curation, investigation, software.

Weizhi Deng: software, visualization.

**Code/Data availability**: All processed data, results, and codes are freely available for download at https://doi.org/10.6084/m9.figshare.10252526.v1 and https://doi.org/10.6084/m9.figshare.12386135.v1

Competing interests: The authors declare that they have no conflict of interest.

## Acknowledgements

We would like to acknowledge the PIs of the MISR, MOPITT, and OMI instruments for providing the remote sensing measurements, and the NCEP and MERRA2 reanalysis project for providing the meteorology and hydrophobic black carbon measurements. MISR data was obtained from https://misr.jpl.nasa.gov/getData/accessData/, MOPITT data was obtained from https://eosweb.larc.nasa.gov/datapool. OMI data was obtained from https://disc.gsfc.nasa.gov/mirador-guide?tree=project&project=OMI, NCEP data was obtained from https://www.esrl.noaa.gov/psd/data/gridded/data.ncep.reanalysis.html, and MERRA data was obtained from https://disc.gsfc.nasa.gov/datacollection/M2I3NVAER_5.12.4.html. The work was supported by the Chinese National Young Thousand Talents Program (Project 41180002), the Chinese National Natural Science Foundation (Project 41030028), and the Guangdong Provincial Young Talent Support Fund (Project 42150003).

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

**Figures**

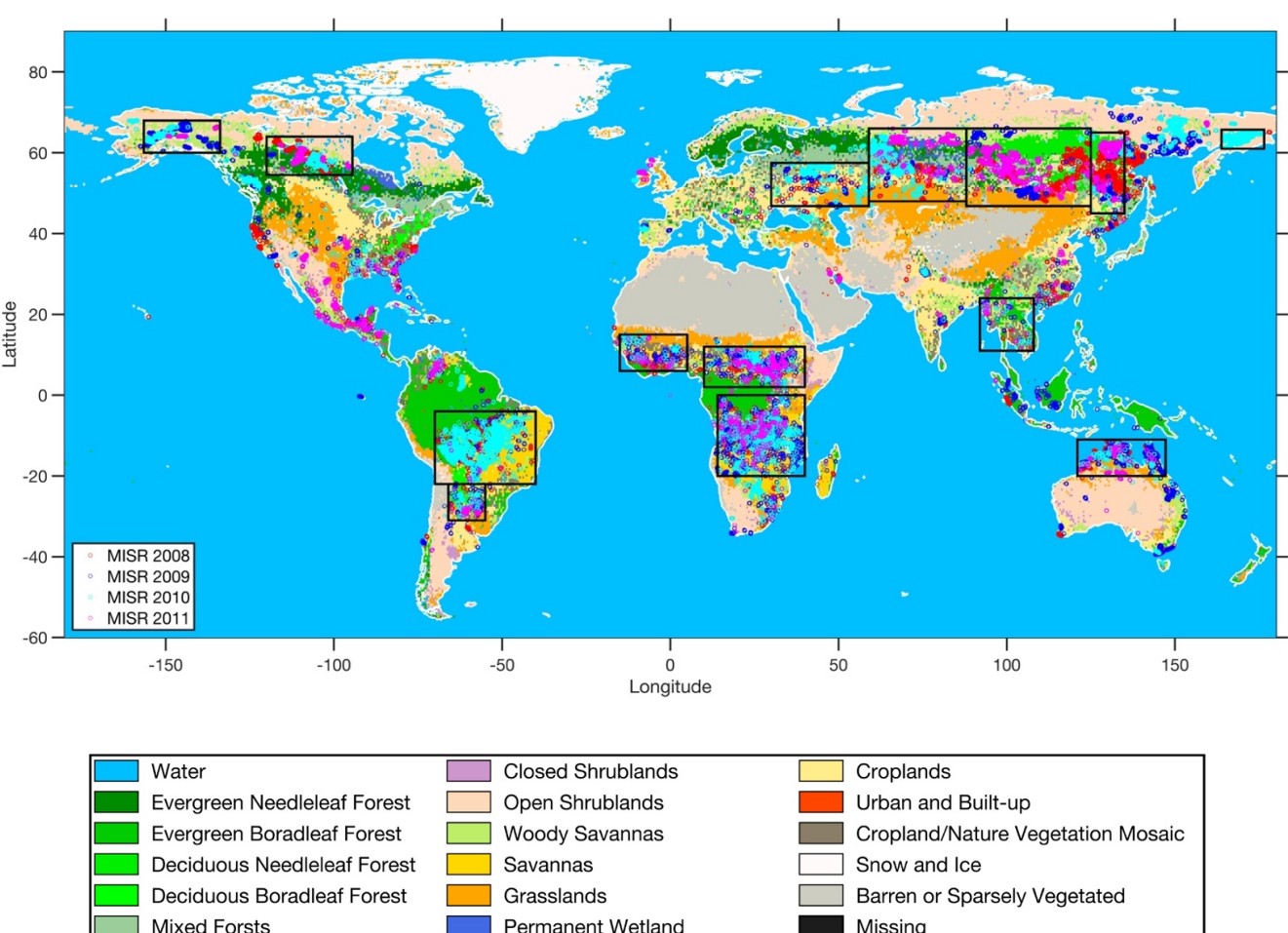

**Figure 1: Land surface type at each of the daily MISR measurements from January 2008 to June 2011. Each dot corresponds to an individual aerosol plume, with different colors representing different years.**

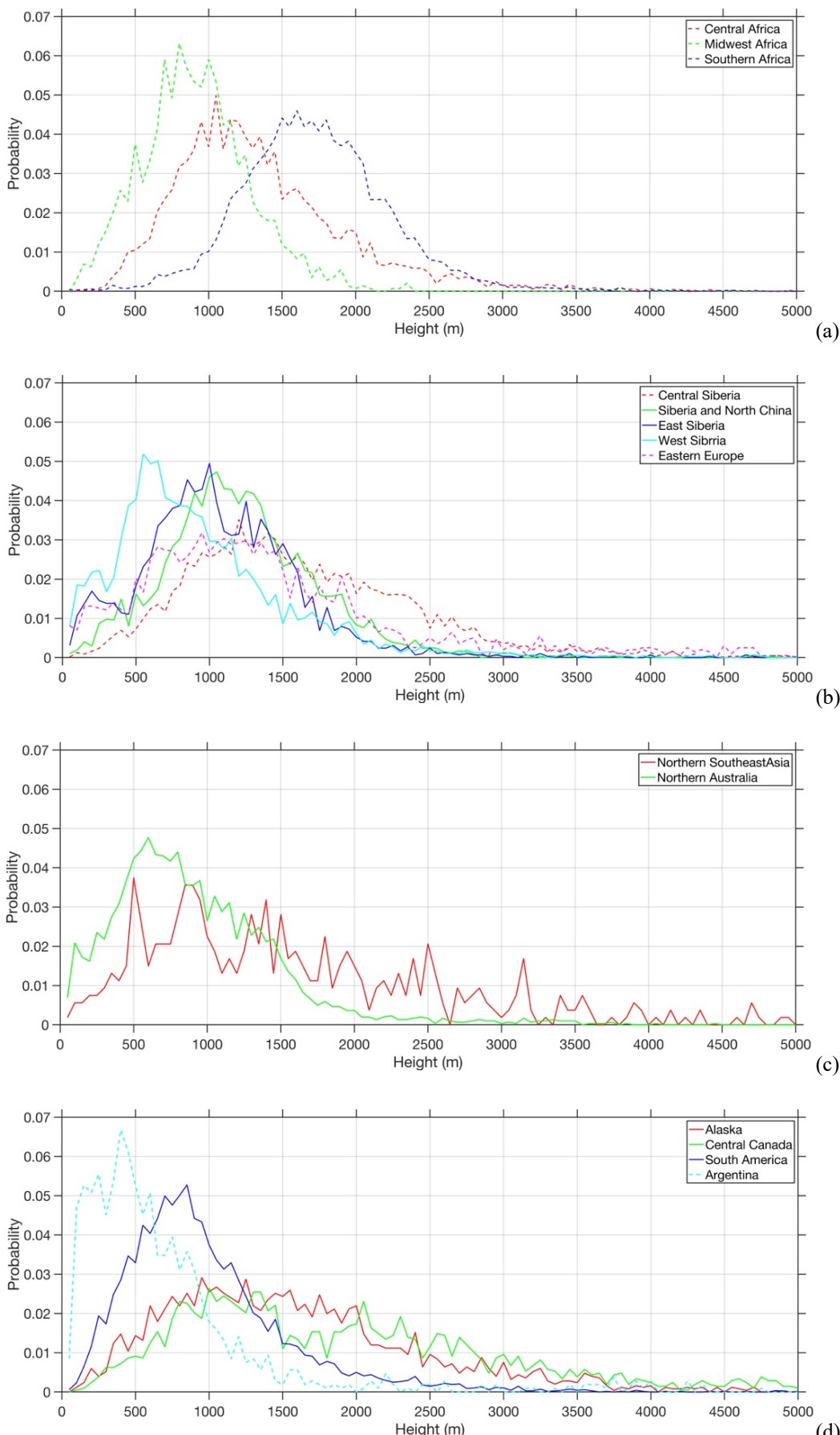

**Figure 2: PDFs of all daily MISR plume height measurements from January 2008 through June 2011 (which are 5000m or less) over each of the following geographic regions: (a) Africa, (b) Eurasian High Latitudes, (c) Tropical Asia, and (d) the Americas. Solid lines correspond to regions which have a successful regression model, while dashed lines are regions which do not.**

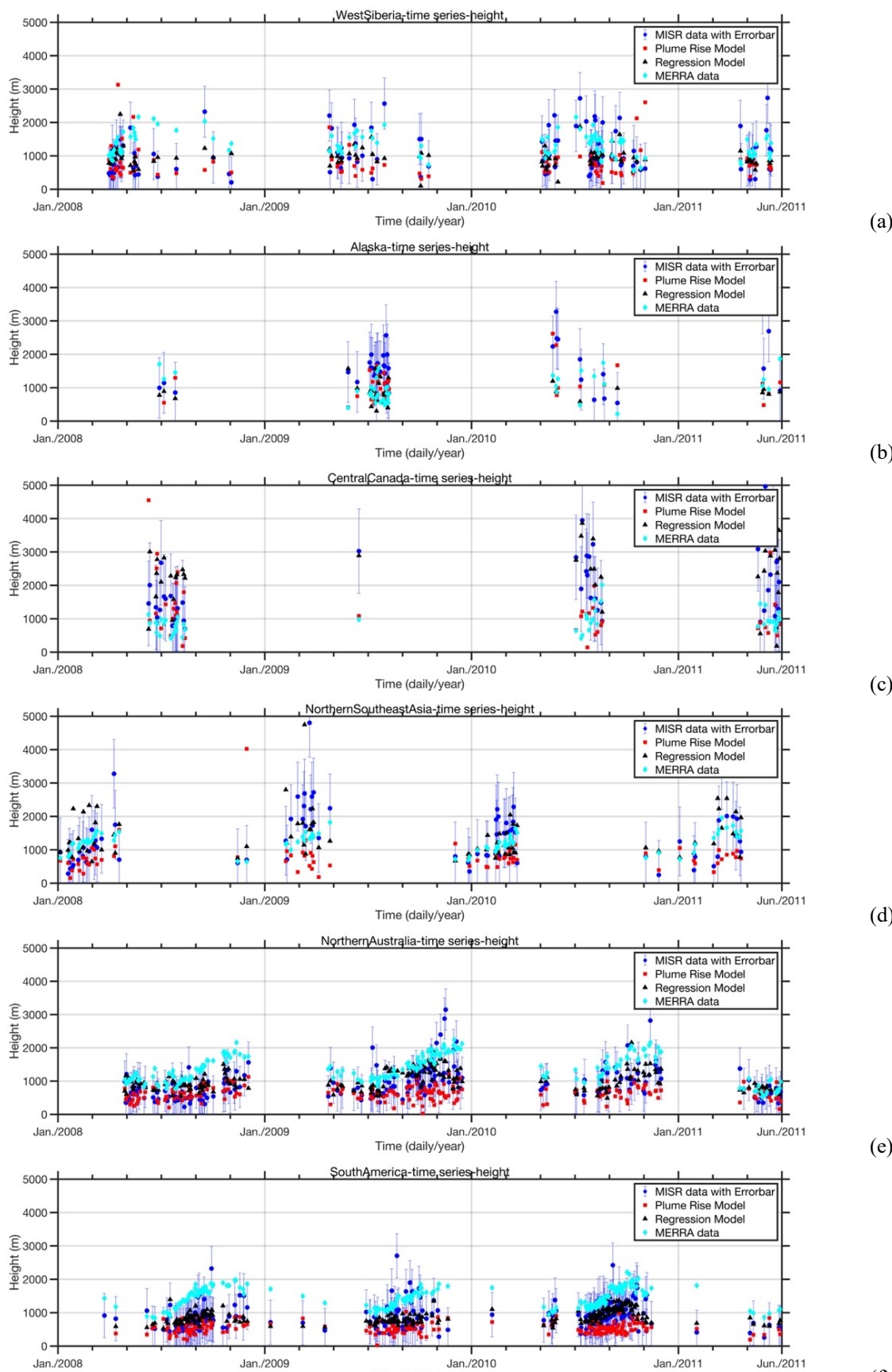

(a)

(b)

(c)

(d)

(e)

(f)

**Figure 3: Time series of daily average measured MISR aerosol height (blue circles [m]) with an error bar corresponding to 1 sigma (blue bars [m]), the plume rise model height (red squares [m]), the regression model height (black squares [m]), and the MERRA hydrophobic black carbon mean height (blue diamonds [m]). Part (a) corresponds to West Siberia, part (b) to Alaska, part (c) to Central Canada, part (d) to Northern Southeast Asia, part (e) to Northern**

**Australia, and part (f) to South America. Missing data points are due to a lack of MISR measurements and/or measurements of regression model predictor(s).**

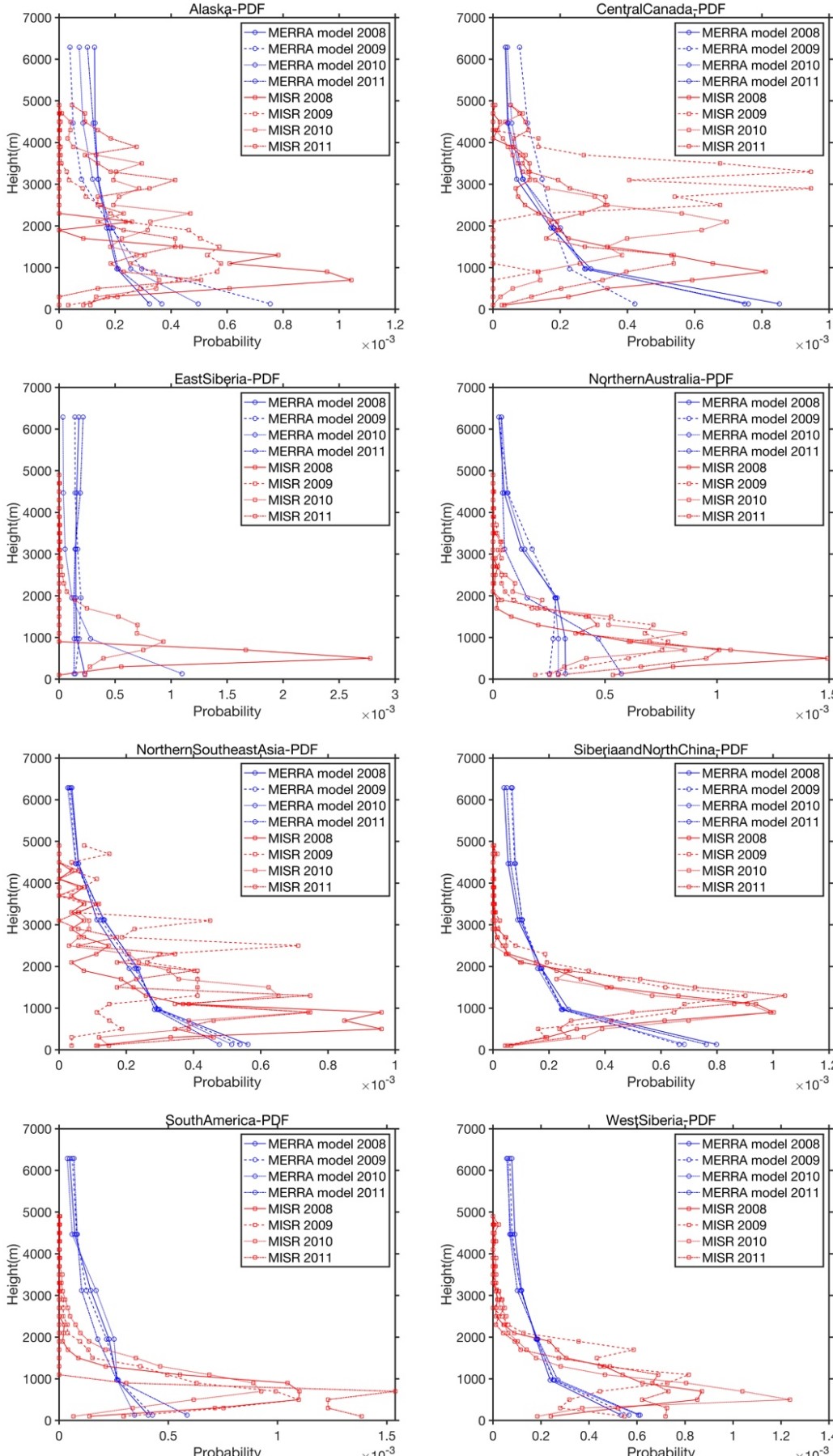

**Figure 4: PDF of the vertical distribution of MISR heights (red lines for 2008, red dashes for 2009, red dots for 2010, and red dash-dots for 2011) and MERRA hydrophobic black carbon heights (blue lines, color scheme the same as for MISR). These plots are only over regions in which the regression model applies.**

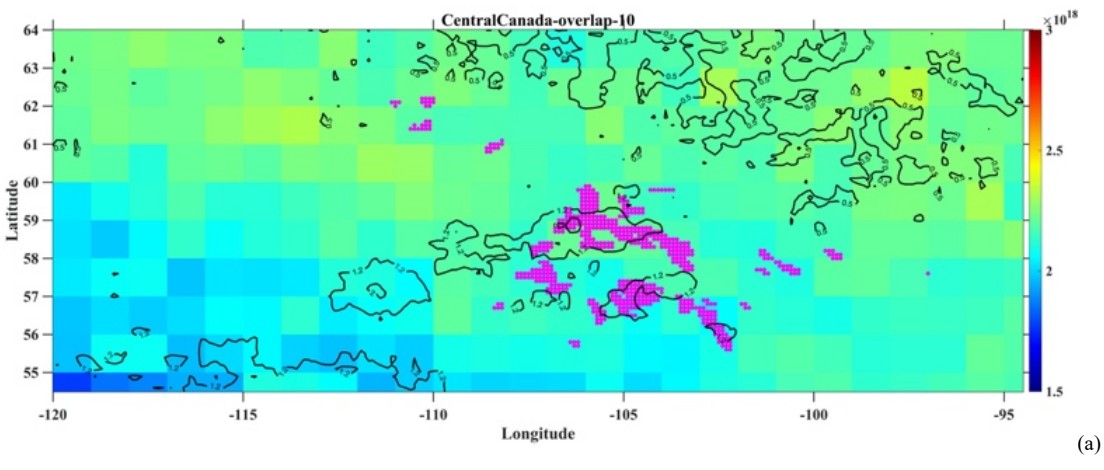

(a)

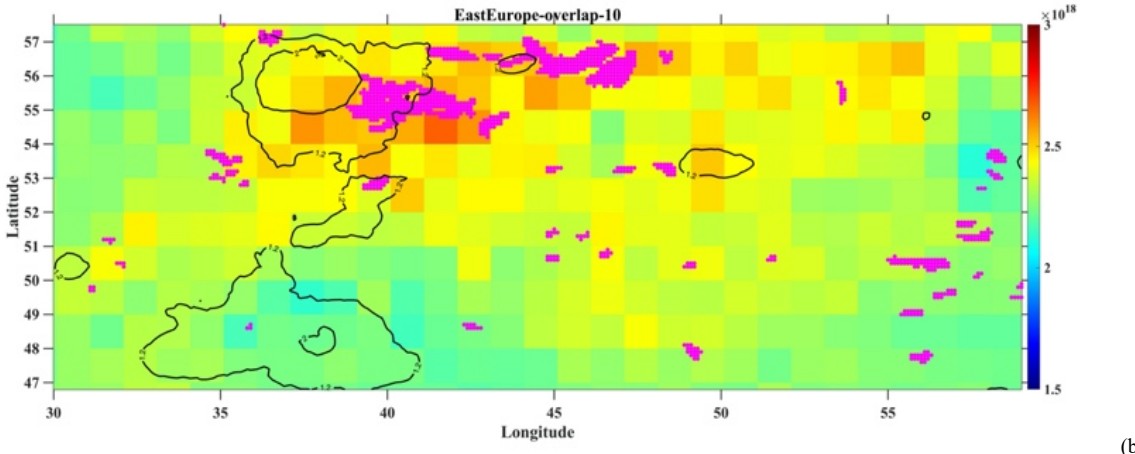

(b)

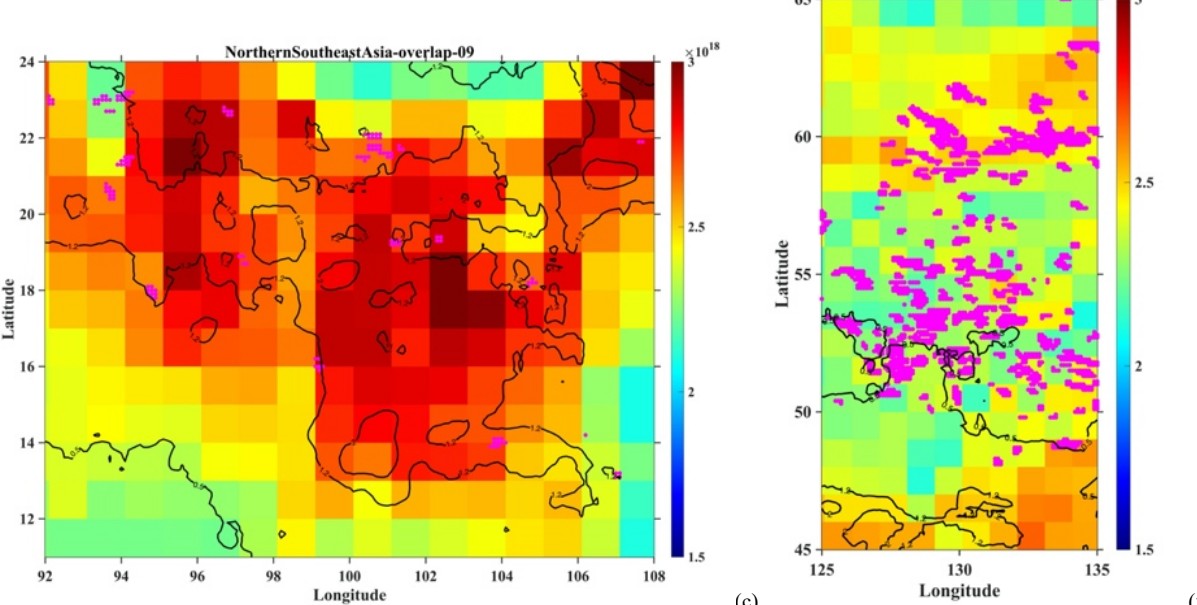

(c)

(f)

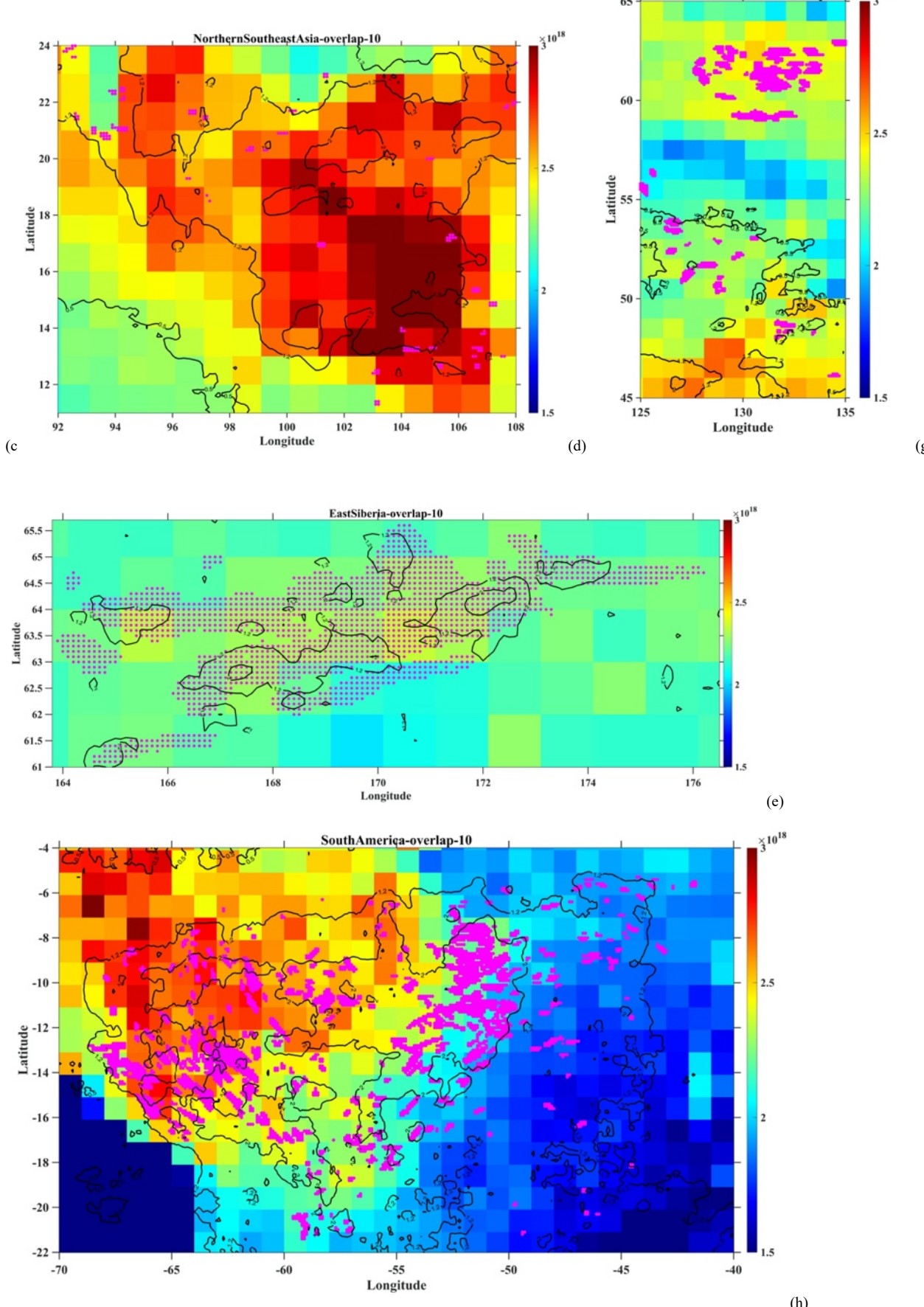

(c)

(d)

(g)

(e)

(h)

**Figure 5: Spatial distribution of annual fires (magenta dots), mean NO₂ column loading on days where there are fires (black isopleths [*10¹⁵ mol/cm²]), and mean CO column loading on days where there are fires (Colorbar, mol/cm²). The corresponding regions are: (a) 2010 Central Canada, (b) 2010 East Europe, (c) 2009 and (d) 2010 Northern Southeast Asia, (e) 2010 East Siberia, (f) 2008 and (g) 2011 Siberia and Northern China, and (h) 2010 South America.**

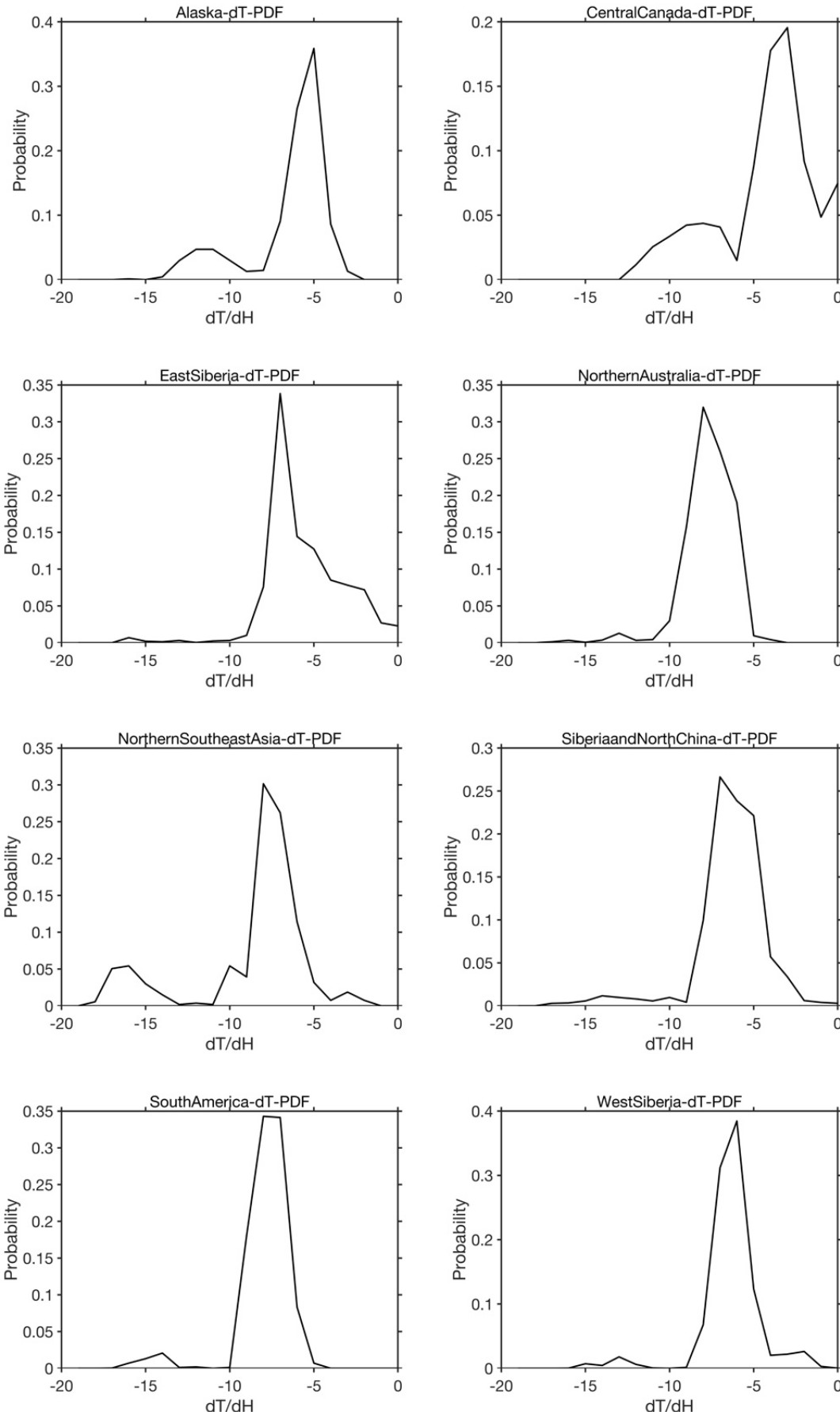

**Figure 6: PDFs of the NCEP reanalysis vertical temperature gradient d[K]/d[km] over the locations and days that contain MISR plumes. The 8 regions over which the regression model are valid are shown.**

**Tables**

| | $C_{NO2}$ (entire box) | $C_{NO2}$ (fire only) | $C_{CO}$ (entire box) | $C_{CO}$ (fire only) |
|---|---|---|---|---|
| Central Africa | 1.36e+15 | 3.24e+15 | 2.24e+18 | 2.49e+18 |
| Midwest Africa | 1.20e+15 | 3.12e+15 | 2.45e+18 | 2.60e+18 |
| Southern Africa | 1.40e+15 | 3.60e+15 | 1.94e+18 | 2.24e+18 |
| Central Siberia | 8.63e+14 | 1.11e+15 | 2.04e+18 | 2.78e+18 |
| Siberia and Northern China | 1.36e+15 | 1.13e+15 | 2.31e+18 | 2.90e+18 |
| Eastern Siberia | 4.74e+14 | 1.50e+15 | 2.25e+18 | 2.38e+18 |
| Western Siberia | 1.21e+15 | 1.70e+15 | 2.20e+18 | 2.52e+18 |
| Northern Southeast Asia | 1.43e+15 | 2.94e+15 | 2.58e+18 | 3.09e+18 |
| Northern Australia | 7.53e+14 | 1.73e+15 | 1.50e+18 | 1.73e+18 |
| Alaska | 7.63e+14 | 1.46e+15 | 2.07e+18 | 2.12e+18 |
| Central Canada | 5.98e+14 | 1.02e+15 | 2.13e+18 | 2.15e+18 |
| South America | 1.16e+15 | 6.36e+15 | 1.78e+18 | 3.08e+18 |
| Argentina | 1.22e+15 | 1.32e+15 | 1.51e+18 | 1.62e+18 |
| Eastern Europe | 1.70e+15 | 1.81e+15 | 2.25e+18 | 2.79e+18 |

**Table 1: Statistical summary of measured column loadings of OMI NO$_2$ [molecule/cm$^2$] and MOPITT CO [molecule/cm$^2$] averaged from January 2008 to June 2011, over each entire boxed region (entire box) as well as the subset in space and time containing active fires (fire only).**

| Region | α | β | γ | δ | ε | $R^2$ |
|---|---|---|---|---|---|---|
| Siberia and North China | 110 | 318 | NaN | 300 | -518 | 0.26 |
| East Siberia | -163 | -657 | 1480 | NaN | 437 | 0.41 |
| West Siberia | 241 | 196 | -221 | NaN | -263 | 0.22 |
| Northern Southeast Asia | 367 | 139 | 912 | NaN | 355 | 0.31 |
| Northern Australia | 211 | -4 | NaN | 1820 | -1580 | 0.24 |
| Alaska | 163 | 18 | 2674 | -892 | NaN | 0.37 |
| Central Canada | -232 | 334 | NaN | 3190 | -1970 | 0.50 |
| South America | 226 | 57 | 314 | NaN | 8 | 0.30 |

Table 2: Best fit values for the various coefficients of the regression models based on Eq.1-7. NaN refers to predictors which are not associated with the given model.

| | MISR data | Plume Rise Model | RMS | Regression Model | RMS | MERRA Data | RMS |
|---|---|---|---|---|---|---|---|
| Central Africa | 1.36 *(0.80)* | 0.59 *(0.22)* | 0.95 | NAN | NAN | **1.72 *(0.50)*** | **0.56** |
| Midwest Africa | 0.90 *(0.42)* | 0.60 *(0.23)* | 0.47 | NAN | NAN | **1.42 *(0.45)*** | **0.41** |
| South Africa | 1.71 *(0.56)* | 0.58 *(0.23)* | 1.18 | NAN | NAN | **1.64 *(0.50)*** | **0.44** |
| Central Siberia | 1.64 *(0.90)* | 0.87 *(0.89)* | 1.01 | NAN | NAN | **2.11 *(1.01)*** | **0.66** |
| Siberia and North China | 1.27 *(0.97)* | 0.80 *(0.64)* | 0.69 | **1.07 *(0.30)*** | **0.42** | 2.06 *(1.20)* | 0.52 |
| Eastern Siberia | 1.12 *(1.00)* | 0.68 *(0.34)* | 0.52 | **1.32 *(0.65)*** | **0.35** | 3.13 *(1.09)* | 0.68 |
| West Siberia | 0.95 *(0.77)* | 0.79 *(0.95)* | 0.67 | **0.97 *(0.29)*** | **0.47** | 1.71 *(0.84)* | 0.53 |
| Northern Southeast Asia | 1.57 *(1.03)* | 0.73 *(0.38)* | 1.04 | **1.42 *(0.51)*** | **0.68** | 1.40 *(0.63)* | 0.75 |
| Northern Australia | 0.90 *(0.62)* | 0.64 *(0.29)* | 0.57 | **1.12 *(0.38)*** | **0.52** | 1.69 *(0.63)* | 0.59 |
| Alaska | 1.57 *(0.91)* | 1.39 *(3.03)* | 0.88 | **1.26 *(0.45)*** | **0.77** | 2.48 *(0.97)* | 1.01 |
| Central Canada | 1.97 *(1.26)* | 1.73 *(2.19)* | 1.36 | **2.13 *(1.72)*** | **1.20** | 2.54 *(1.17)* | 1.36 |
| South America | 0.97 *(0.66)* | 0.50 *(0.21)* | 0.52 | **0.95 *(0.22)*** | **0.37** | 1.92 *(0.91)* | 0.60 |
| Argentina | 0.69 *(0.70)* | **0.65 *(0.25)*** | **0.40** | NAN | NAN | 1.30 *(0.49)* | 0.52 |
| Eastern Europe | 1.41 *(1.05)* | 1.27 *(2.67)* | 0.85 | NAN | NAN | **1.15 *(0.59)*** | **0.65** |

Table 3: Statistics of measured MISR plume heights and (standard deviations) (2nd column [km]) using all available daily data from Jan 2008 to Jun 2011; plume rise model heights and (standard deviations) (3rd column [km]); RMS error between the MISR plume heights and plume rise model heights (4th column [km]); regression model heights and (standard deviations) (5th column [km]); RMS error between the MISR plume heights and regression model heights (6th column [km]); MERRA daily mean hydrophobic black carbon heights and (standard deviations) (7th column [km]); and finally the RMS error between the MISR plume heights and MERRA daily hydrophobic black carbon heights (8th column [km]). NaN indicates that the regression model failed over the respective region. The model type with the lowest RMS error over each region is given in "Bold".

|  | MISR 10% | MISR 30% | MISR 50% | MISR 70% | MISR 90% | PRM 10% | PRM 30% | PRM 50% | PRM 70% | PRM 90% |
|---|---|---|---|---|---|---|---|---|---|---|
| Central Africa | 0.70 | 0.99 | 1.22 | 1.53 | 2.10 | 0.33 | 0.47 | 0.57 | 0.68 | 0.85 |
| Midwest Africa | 0.43 | 0.69 | 0.87 | 1.05 | 1.37 | 0.30 | 0.49 | 0.60 | 0.70 | 0.85 |
| South Africa | 1.12 | 1.44 | 1.67 | 1.92 | 2.31 | 0.32 | 0.46 | 0.56 | 0.67 | 0.84 |
| Central Siberia | 0.75 | 1.15 | 1.48 | 1.93 | 2.62 | 0.38 | 0.59 | 0.74 | 0.91 | 1.27 |
| Siberia and North China | 0.58 | 0.92 | 1.15 | 1.41 | 1.88 | 0.38 | 0.55 | 0.68 | 0.84 | 1.24 |
| East Siberia | 0.41 | 0.77 | 1.00 | 1.29 | 1.69 | 0.36 | 0.49 | 0.62 | 0.78 | 0.97 |
| West Siberia | 0.28 | 0.56 | 0.79 | 1.09 | 1.71 | 0.38 | 0.52 | 0.62 | 0.76 | 1.14 |
| Northern Southeast Asia | 0.48 | 0.87 | 1.35 | 1.91 | 3.03 | 0.32 | 0.55 | 0.71 | 0.84 | 1.10 |
| Northern Australia | 0.28 | 0.56 | 0.79 | 1.09 | 1.52 | 0.34 | 0.49 | 0.63 | 0.75 | 0.93 |
| Alaska | 0.59 | 1.02 | 1.43 | 1.88 | 2.78 | 0.52 | 0.83 | 1.00 | 1.20 | 1.56 |
| Central Canada | 0.72 | 1.16 | 1.73 | 2.36 | 3.51 | 0.51 | 0.74 | 0.98 | 1.68 | 3.04 |
| South America | 0.38 | 0.64 | 0.85 | 1.11 | 1.65 | 0.26 | 0.39 | 0.50 | 0.60 | 0.77 |
| Argentina | 0.14 | 0.34 | 0.51 | 0.75 | 1.26 | 0.34 | 0.50 | 0.63 | 0.76 | 0.97 |
| East Europe | 0.44 | 0.85 | 1.19 | 1.60 | 2.63 | 0.47 | 0.64 | 0.82 | 1.08 | 1.97 |

(a)

|  | RM 10% | RM 30% | RM 50% | RM 70% | RM 90% | MERRA 10% | MERRA 30% | MERRA 50% | MERRA 70% | MERRA 90% |
|---|---|---|---|---|---|---|---|---|---|---|
| Central Africa | nan | nan | nan | nan | nan | 1.08 | 1.47 | 1.71 | 1.96 | 2.33 |
| Midwest Africa | nan | nan | nan | nan | nan | 0.87 | 1.18 | 1.40 | 1.62 | 1.99 |
| South Africa | nan | nan | nan | nan | nan | 1.01 | 1.35 | 1.62 | 1.90 | 2.29 |
| Central Siberia | nan | nan | nan | nan | nan | 0.87 | 1.51 | 1.99 | 2.53 | 3.49 |
| Siberia and North China | 0.89 | 1.02 | 1.13 | 1.27 | 1.50 | 0.55 | 1.27 | 1.92 | 2.64 | 3.74 |
| East Siberia | 0.95 | 1.41 | 1.66 | 1.88 | 2.66 | 1.72 | 2.57 | 3.14 | 3.72 | 4.56 |
| West Siberia | 0.72 | 0.84 | 0.93 | 1.03 | 1.22 | 0.67 | 1.22 | 1.63 | 2.06 | 2.81 |
| Northern Southeast Asia | 0.81 | 1.00 | 1.20 | 1.69 | 2.64 | 0.68 | 0.99 | 1.29 | 1.65 | 2.29 |
| Northern Australia | 0.71 | 0.87 | 1.04 | 1.25 | 1.53 | 0.91 | 1.29 | 1.64 | 2.01 | 2.52 |
| Alaska | 0.30 | 0.80 | 0.82 | 0.85 | 1.35 | 1.25 | 1.94 | 2.43 | 2.94 | 3.76 |
| Central Canada | 0.80 | 2.01 | 2.28 | 2.78 | 4.59 | 1.02 | 1.81 | 2.49 | 3.22 | 4.13 |
| South America | 0.71 | 0.86 | 0.98 | 1.11 | 1.36 | 0.90 | 1.38 | 1.77 | 2.22 | 3.19 |
| Argentina | nan | nan | nan | nan | nan | 0.70 | 1.01 | 1.25 | 1.52 | 1.94 |
| East Europe | nan | nan | nan | nan | nan | 0.43 | 0.78 | 1.09 | 1.40 | 1.90 |

(b)

**Table 4: Statistics of the 10%, 30%, median, 70% and 90% percentile heights [km] of MISR heights and plume rise model heights (a), and regression model heights and MERRA heights (b). NaN refers to regions where there is no regression model result.**