# Peer review of "Constraining the relationships between aerosol height, aerosol optical depth and total column trace gas measurements using remote sensing and models"

_Atmospheric Chemistry and Physics, 2019_

## Referee Comment (RC1) · Anonymous Referee #1 · 18 Mar 2020

Summary-

The paper compares a simple plume model and a multiple linear regression (MLR) model approach to observed plume heights from MISR. The plume model and MLR models use overlapping data sets to predict plume height. The authors find that the plume model generally under performs the MLR models.

The use of overlapping data to train the MLR models and to get predictions from the plume model is interesting. However, the use of a single plume model from 1965

is poorly motivated. The authors need to discuss in detail the current state of the field in plume modelling (which I feel must have progressed somewhat in the past 50 years) and compare several plume models to the MLR models. At some level the MLR model will always get a better agreement with data because mathematically it is going to always minimize unexplained variance, in contrast to the plume model, which is based on some physical understanding. This overfitting problem could be solved by training the MLR model in one region and applying it to other regions. The authors also train 7 MLR models based on a combination of different predictors. The way that this feeds into the comparisons between the 'regression model' and plume model is poorly described. The authors need to either use all the predictors, or come up with some objective methodology to throw out some (eg machine learning).

The paper seems rushed and has many grammatical errors. The number of figures must be increased to make it clearer what the analysis shows. The statistical analysis is unclear and in some cases contradictory and arbitrary (the authors describe predictors as orthogonal and then include a predictor that is a ratio of other predictors, data that agrees too poorly is thrown out).

There is no comparison of these results to any sort of reasonable chemical transport model (for instance MERRA2 might even have sufficient data to tell us about plume height and would be a more fair comparison).

Because I feel that the amount of work to add additional plume models, make the regression analysis more objective, and incorporate some chemical transport modelling results requires more work than can be accomplished in a review period I recommend rejection. ———- L18 Just saying the MLR model does a better job is a bit disingenuous. Linear least squares will always maximize variance explained. The authors need to show that they do some sort of out of sample testing.

L32 Use of significant should be reserved for statistical statements. Consider using 'substantial'.

L34 'and are known'

L35 I believe biomass burning is also emitted at the surface and you mean it is moved into the upper atmosphere.

L40 The statement that aerosols above the PBL have a bigger influence on the atmosphere may be true in some context, and the authors do provide citations, but they need to be a bit more specific here. I assume they mean in some sort of normalized sense (eg Pinatubo had a big influence on global mean temperature, but in an integrated sense aerosol in the boundary layer probably has a bigger impact). Either way, while a very interesting point to make, the authors might want to expand on this statement a bit for clarity.

L45 Who used? I think the authors have a typo and all the citations have stuck together.

L53 Lidar isn't capitalized: https://www-calipso.larc.nasa.gov/

L81 Large majority is redundant

L99 typo, remove 'the'

L144 Specifically

L145 Does this mean that when you have cloud or aerosol you don't get CO measurements?

L156 NO2 also has substantial industrial sources. The way that this is written implies that NO2 is only from fires.

L187 Note that inputs are not necessarily orthogonal, unless you pretreat inputs somehow. For example, NO2/CO is going to be correlated with NO2 and CO.

L188 Typo in this sentence.

L216 This sentence is very unclear- how are you 'injecting additional information'? As you say earlier all data sets have to be present. This seems to imply that data

points with missing data will sometimes be considered and additional information will sometimes be 'injected'.

L218 It is also unclear how you intend to reduce bias. Do you mean that you will try out data sets that measure the same quantity to get an estimate of bias.

L254 It would be good to define FRP somewhere in the intro or methods in terms of its physics (for people outside the biomass burning community).

L270 something that I think needs to be discussed in the use of this plume rise model is that it is based on a model from 1965. In the methods there need to be a few sentences on why this model has not been improved upon since then, or why it is an appropriate comparison to the MLR model. Not discussing this runs the risk of making the plume model seem like a straw man to those outside the plume modelling community. Another aspect of this plume rise model is that earlier the authors state that it begins to fail for small fires. The analysis should really be subset to fires that satisfy the assumptions going into the model, rather than degrading the model with fires that the plume rise model is not designed for.

L329 A citation to a review article here might be helpful.

L332 Different than each other? Do you mean when the plume model and the measurements? If this is the case this also seems fairly arbitrary to be testing the model and throwing out the results when they are poor.

L340 Is this just a function of bias from the plume rise model treating fires that are smaller and thus don't satisfy assumptions in the model?

L341 how well the data what?

L344 While I understand the attraction of minimizing the number of figures, but this article only has 3 in the main text. I feel that the PDFs of modeled and observed plume heights could be moved to the main text.

L365 How does the analysis account for times when the area is very crowded with burning? How does it tell where plumes actually originate from? Can a plume from another fire be mistagged or affect plumes from a nearby fire?

L375 A clear list of assumptions in the methods would be good. I assume there is more than one plume rise model in the literature (for example https://link.springer.com/article/10.1007/s10661-005-1611-y). The authors must show results from at least two leading plume rise models to show that the poor results of the 1965 model are not just due to poor construction of the model and limitations in what it can do (and applying the model outside of its assumed conditions).

L385 Is this because Argentina is dominated by the Pampas and fires tend to be over large areas and are uniform and the meteorology is relatively less complex?

L397 I think rather than coming up with 7 combinations of predictors a better approach might be to only have one model with all the predictors or use some sort of objective algorithm (eg machine learning) to remove low explained variance predictors. Arbitrarily coming up with 7 models seems like it will almost always guarantee a model works well.

L408 Fragment

L411 Again, I don't understand how this is an evaluation if predictions that agree too poorly are removed.

L430 The three regions shown in Fig 3 are for a few plumes (judging by plotted data points) and for only a subset of the plumes in Fig1.

L478 Which of the regression models is the new method?

L483 What are the 'modelled results' in contrast to the plume and regression models?

L497 Somewhere there needs to a scatter plot of MLR model plume height versus observations. One possibility is that you are just fitting the mean. The MLR model

is guaranteed to do this well (it minimizes unexplained variance). To do this correctly you should train the model on one region and apply it to other regions to get rid of the overfitting problem.

Fig1 I am not sure how useful this plot is because the dots obscure the land surface type.

Fig2 Please use some different line styles and markers. Most of these colors are indistinguishable.

---

## Referee Comment (RC2) · Anonymous Referee #2 · 20 Apr 2020

Review of Wang et al.:

I will keep this short and to the point. I think the basic idea of trying to investigate the relationship between trace gas/aerosol plume height and the pollutant loading is good. But having read the manuscript few times, I do not believe the authors have approached the problem with the right tools. My opinion/review is mostly from the observational perspective and I don't know much about the plume models.

1) Why use the total column values of NO2 and CO, when the authors themselves

show how, depending on the region, aerosols can be lifted to different heights. What do we actually scientifically gain by looking at the total column only? It is not a surprise that when episodes of strong pollution occur (e.g. fires, biomass burning), the total column values will increase and depending on the thermodynamical conditions (e.g. strength of convection) the lofting will occur. I understand that the vertically resolved observations of NO2 are not available, but altitude-resolved CO retrievals are available from a number of sensors, MOPITT, AIRS, IASI etc. I also wonder why the authors don't use aerosol layer heights from CALIPSO (possibly combined with OMI)? Wouldn't that be the most accurate account of plume heights?

2) The lifetimes of CO and NO2 are very different. CO has much more homogenized distribution in the atmosphere, especially as the altitude increases due to transport processes etc. So can the authors disentangle this background signal from the one that is associated with the biomass burning plumes for CO, especially over those regions that already have strong background variability in industrial+traffic pollution?

3) There is virtually no description of how different satellite data products are quality controlled, analysed etc. The devil is in the details. What quality flags are used? How are cloudy/non-cloudy cases handled? Is there a consistency in such cases across all datasets? How is the sampling affected by the quality control?

―――――――――――――――――

---

## Author Response (AR1)

Dear Editor, Reviewer 1, and Reviewer 2:

Thank you for taking the time to provide all of your meaningful and insightful comments and suggestions. We have taken them all into serious consideration and have strived to work hard to address them all. In this response, your original comments are given in yellow highlight, our responses are given in blue highlight and updates to the paper are given in pink highlight. We respond to Reviewer 1 first in full, and then respond to Reviewer 2 in full. If answers are given above to a previous question, we may refer the reader to "see above" or something similar. Thank you again for your time and deep insights!

Response to Author 1:

Summary-
The paper compares a simple plume model and a multiple linear regression (MLR) model approach to observed plume heights from MISR. The plume model and MLR models use overlapping data sets to predict plume height. The authors find that the plume model generally under performs the MLR models. The use of overlapping data to train the MLR models and to get predictions from the plume model is interesting. However, the use of a single plume model from 1965 is poorly motivated. The authors need to discuss in detail the current state of the field in plume modelling (which I feel must have progressed somewhat in the past 50 years) and compare several plume models to the MLR models. At some level the MLR model will always get a better agreement with data because mathematically it is going to always minimize unexplained variance, in contrast to the plume model, which is based on some physical understanding.

This is an essential and important part of the paper that we have made modifications to make clearer. Thank you for pointing out this essential communication issue!

One of the critical assumptions of all plume rise models is that the vertical rise is controlled by the buoyancy and vertical motion forces. The input from the fire is the heat co-emitted with the aerosols and gasses. Any initial vertical momentum applied at the fire start point and the atmospheric temperature distribution are a function of the atmospheric state. As the heated air from the fire rises through the atmospheric column, it interacts with the background conditions and eventually an equilibrium state is reached.

However, there are a few factors which have been found to be important, but are missing in this approximation. There are now more than a few papers (Guo et al., 2019; Tao et al., 2012; and Mims et al., 2010) that show the aerosols co-emitted with the heat absorb and scatter a significant amount of incoming solar radiation in the daytime and outgoing IR radiation in the nighttime, changing the energy structure of the column above and below the point at which the aerosols are located in the vertical, as well and the buoyancy of the air parcel containing the aerosols. Secondly, in the case where there is a large-scale aerosol cloud due to extensive burning over a significant land surface area, this widely distributed cloud of aerosols in the atmosphere further changes the absorption and scattering of the atmosphere at the meso-scale (Wang et al., 2009; Ekman et al., 2011; Cohen et al., 2011), in turn further changing the atmosphere's general energy balance. A third issue is the radiative-convective equilibrium occurring within the column over which the air parcels rise also depend on the loadings of clouds and aerosols above and below the parcel of interest. Therefore, any physically-based plume-rise model, as currently found in the literature and used by the modeling communities, regardless of whether it was fitted 50 years ago or has been slightly improved in terms of its coefficients under different conditions, still cannot capture the required

set of physics to be fully realistic. Hence, we do not feel that the issue is how long ago the currently used theory was developed is overly relevant. In fact, we attempt to form a regression model (as you term "MLR model" or simply "MLR" from this point forward) specifically to cater to this assumption (more on this later).

The following paragraph has been added into the paper in Section 3

Third, the range of the seven regression models is an attempt to intelligently account for the fact that the column loadings of the CO and $NO_2$ offer physical meaning and insight, as compared to merely being an attempt to minimize any unexplained variance. We argue that the column values of both CO and $NO_2$ are both directly and indirectly related to the magnitude and the height of the vertical aerosol column. Due to the fact that the emissions of $NO_2$ is a strong function of the fire temperature, and its short atmospheric lifetime, the $NO_2$ is strongly related to the temperature of the fire, or the FRP, which is one of the essential driving forces of the buoyancy. This issue is strongly coupled with the fact that FRP is also one of the most error-prone of the measurements commonly used to drive the plume-rise models, with the FRP commonly underestimated in the tropics due to clouds and aerosols, as given in Kaiser et al. (2012), Cohen et al. (2018), and Lin et al. (2020a). Additionally, the amount of CO produced is a function of the total amount of biomass burned as well as the wetness of the surface itself where the burning occurred, and hence the CO column loading is also physically related to the properties of the fires. In fact, using a measure of the CO column can help us to overcome the physical constraints that current measurements have in terms of addressing the issues of how much peat or understory has burned, or if such fires which are occurring without direct line of sight from above can even be detected by the current fire detection processes at all (Leung et al., 2007; Ichoku et al., 2008). The combination of high $NO_2$ (which is more produced at higher temperature) and low CO (which is more produced at higher temperature) means that the ratio of $NO_2$ to CO also provides further physical insight into the non-linearities associated with the fire temperature, wetness, and possibility of other heat sources/sinks at the fire/atmosphere interface such as smoldering, conversion to latent heat, etc.

This overfitting problem could be solved by training the MLR model in one region and applying it to other regions. The authors also train 7 MLR models based on a combination of different predictors. The way that this feeds into the comparisons between the 'regression model' and plume model is poorly described. The authors need to either use all the predictors, or come up with some objective methodology to throw out some (eg machine learning).

This is an interesting point, and I believe worthwhile for follow-up work. It is not well known if such a single model would allow for a single idealized modeling format to be achieved throughout the entire real world for three different reasons. First, the biomass type and loading are different across different regions of the world. Secondly, the climatology of the soil moisture, boundary layer, and the free atmospheric vertical profile are also not consistent across different parts of the world. Finally, these different regions are sometimes impacted by human emissions and sources of co-emitted heat, aerosols and gasses, and sometimes not. For this reason, in this work we are focusing first and foremost on the idea that applying a physically based MLR model can give us insights, and to figuring out where such approach may add value for the community as a whole.

The results show clearly that the versions of the regression model that best model the height all have the $NO_2$ term in them. Furthermore, over all regions except for one, the best fitting regression models also have a term representing CO. Therefore, the number of regression models computed, in retrospect, could have been reduced, with models 5, 6, and 7 excluded. This end result shows clearly that the simpler plume

rise regression model representation is never superior in any case. Further work could look into how more advanced modeling perspectives may or may not improve upon the framework introduced here. We believe that there is already considerable value and uniqueness offered by this approach.

The following paragraph has been added into the paper in section 3.3

The regression model solely containing $NO_2$ is an approximation of the concept that the heat of the biomass burning should have an important role to play in terms of the plume height. Furthermore, using $NO_2$ in this way helps to get around the inherent underestimation of FRP. The regression model solely containing the CO is a proxy for the concept that the mass of biomass burned should make an important contribution towards the plume height. Inclusion of the CO term is also a way to get around the underapproximation of the total burned area, or of any significant contribution from underground burning.

The following sentences have been added into the paper in section 3.3

The regression model with the non-linear combination of the two is a proxy for the argument that it is the ratio of the heat to the total biomass burned that is an essential physical consideration to take into effect. Furthermore, this final case provides some weight to the concept that a small change in the vertical column concentration may have a stronger than linear effect, as is evidenced by (Ichoku et al., 2008; Zhu et al., 2018), such as in terms of absorbing aerosols (which are themselves produced more so under hot or oxygen starved conditions) in the vertical column altering the ultimate vertical distribution.

The following paragraph has been added into the paper in Section 2.7

The 7 different regression models were chosen so as to cover the entire combination of different ways to fairly and uniformly incorporate the CO and $NO_2$ measurements as well as their underlying physical meanings. The 7th regression model is the approximation of the Plume Rise Model. The 4th and 5th regression models are the approximations of the single-species linear impact of $NO_2$ and CO respectively. The 6th regression model approximates the single-species non-linear impact of $NO_2$ and CO in tandem. Finally, the 1st through 3rd regression models are the approximations of the combination of CO and $NO_2$ in tandem with both linear (model 1), or with one linear and one non-linear combination (models 2 and 3). This approach is consistent with and follows from some of the earlier works which tries to use advanced learning to understand some higher order, simple non-linear forcings, still based on some physical consideration, i.e. Cohen and Prinn, 2011.

The following sentence has been added into the paper in Section 4

As we have demonstrated, the impact of $NO_2$ (as a proxy for the burning temperature) is always essential, and the impact of CO (as a proxy for the total biomass burned) is usually essential as well. We further have shown that the simplest regression model, the approximation of the Plume Rise Model, never yields the best fit to the data.

The paper seems rushed and has many grammatical errors. The number of figures must be increased to make it clearer what the analysis shows.

We have included 3 new figures (Figures 4, 5, and 6) in the paper and 2 new figures in the supplement (Figure S5 and Figure S6). We also have expanded the information provided in Figure 3. Finally, we have added in a new table (Table 4).

Changes made to the spelling and grammar are clearly shown in the track-changes version of the text itself.

The statistical analysis is unclear and in some cases contradictory and arbitrary (the authors describe predictors as orthogonal and then include a predictor that is a ratio of other predictors, data that agrees too poorly is thrown out).

145

As discussed above, the ratio predictor has its own unique physical meaning, describing the ratio of the temperature to the amount of biomass burned at the instantaneous point and time where the burning occurred. The pure $NO_2$ term is also an instantaneous term, describing the temperature of the burning at

150

the time of burning. This is consistent with the fact that the lifetime is $NO_2$ is very short, lasting far less than the day-to-day gap between the measurements. The pure CO term on the other hand is not an instantaneous term, instead describing the total amount of biomass burned over the past day (or days in the case of missing data) between the prior measurement and the most recent measurement. This result is also consistent with the long lifetime in-situ of CO, lasting from weeks to months, as described in Lin et

155

al., 2020a, 2020b. Ideally for future work, we can find a third completely independent measurement which can also provide us a similar piece of knowledge such as provided by the term $[NO_2]/[CO]$, however such may not be possible until the next generation of satellite products is released to accomplish such a goal (i.e. Qin et al., 2020).

160

We agree in full about providing the full set of data over all of the areas. To better facilitate this, we have included more data in **Figure 3 and Figure S3**, including in regions where neither the egression model or plume rise model are found to be good fits. We also have included some extra discussion of these points. Furthermore, we have included an analysis of the black carbon height based on the mean daily MERRA hydrophobic black carbon values on the same days corresponding to where we have MISR height

165

measurements. The data is provided in **Figure 3 (a)-(f)**.

The following Figure has been added as Figure 3

[Figure]

(a)

(b)

170

[Figure]

(c)

(d)

(e)

(f)

**Figure 3: Time series of daily average measured MISR aerosol height (blue circles [m]) with an error bar corresponding to 1 sigma (blue bars [m]), the Plume Rise Model height (red squares [m]), the regression model height (black squares [m]), and the MERRA hydrophobic black carbon mean height (blue diamonds [m]). Part (a) corresponds to West Siberia, part (b) to Alaska, part (c) to Central Canada, part (d) to Northern Southeast Asia, part (e) to Northern Australia, and part (f) to South America. Missing data points are due to a lack of MISR measurements and/or measurements of regression model predictor(s).**

Secondly, we have computed the statistics of the 10%, 30%, median, 70%, and 90% percentile heights of the daily MERRA hydrophobic black carbon heights on the same days where there are also MISR measurements. These results have been combined with the computed the statistics of the 10%, 30%, median, 70% and 90% percentile heights of the MISR measurements, the Plume Rise Model results, and the regression model results, into an expansion of **Table 3** and the new **Table 4**.

The following Paragraph has been added into the paper in Section 3.5

[revised manuscript text omitted]

There is no comparison of these results to any sort of reasonable chemical transport model (for instance MERRA2 might even have sufficient data to tell us about plume height and would be a fairer comparison).

In terms of the RMS error of the mean height over the entire time period, we determine that the MERRA model performs more poorly than the regression model at all places where the regression model passes the test of reliability. We also note that the MERRA RMS error is lower at the locations where the regression model does not pass the reliability test than over regions where it does the pass reliability test. This interesting result may further strengthen the idea that the regression model is accounting for some aspect of non-linearity which the underlying model used for MERRA is not accounting for.

MERRA performs better than the plume rise model in 8 regions, worse in 5 regions, and similarly in 1 region. Again, it is interesting to note that the region where the plume rise model works better than MERRA that does not also work for the regression model is in Argentina. Therefore, in general, these results show that the plume rise model almost never adds value, as compared to MERRA or the Regression approach, except for in Argentina. In the case of Argentina, MERRA has an obvious high bias, possibly due to the effect of the Andes Mountains being a dominant feature over much of this region's total area, and the known problems of global-scale models in representing highly mountainous regions.

Because I feel that the amount of work to add additional plume models, make the re- egression analysis more objective, and incorporate some chemical transport modelling results requires more work than can be accomplished in a review period I recommend rejection.

L18 Just saying the MLR model does a better job is a bit disingenuous. Linear least squares will always maximize variance explained. The authors need to show that they do some sort of out of sample testing.

We believe that the explanations above and comparisons with the Plume Rise Model and MERRA show that the MLR model does a better job. We understand clearly the concept of out of sample testing, but believe that it is not required in the case where, we are training against MISR and comparing against MERRA, that it is not required. We are not using the same dataset for training and comparison. Recall that as a data assimilation product, MERRA should be based on information which is quite different from the MISR plum heights, $NO_2$, and CO used in the training and comparisons.

The following sentences have been added to section 4

Our results show clearly that where we can successfully form a regression model, that it performs better than both the plume rise model and MERRA. The specific forms of the regression model that are the best are those which have NO2 or a combination of NO2 and CO (in particular when the non-linear term NO2/CO is considered). These results are consistent with our hypothesis and literature review that show new forms of non-linearity relating plume rise height to factors influencing buoyancy, radiative transfer, and energy transfer in-situ, and/or biases in remotely sensed measurements of FRP and land-surface products are important. Such are not considered in the present generation of plume rise models (including the global-scale models underlying MERRA). In the cases where we cannot form a regression model, we find that MERRA performs better than the plume rise model everywhere, except for Argentina, which has a unique high mountain just upwind in the Andes, coupled with a very low overall height, all of which are disadvantages for the models underlying MERRA. In general, this shows that improved

model complexity and data assimilation doe produce a better result, as expected.

We propose the results as a first step of a new approach to parameterization that my help us to move forward in terms of improving our ability to reproduce heights of fire plumes for regional and global scale modeling and analysis studies over many different periods of time. We believe that our sample dataset is currently not sufficiently long to form an ideal fit, and hence thought that excluding data to self-compare was not an ideal use of the very limited resources we had. We do hope that as more new datasets are released, the community will have access to more relevant input data, and as more MISR plume height data is released, the community will have more access to better understand the vertical distribution of height.

L32 Use of significant should be reserved for statistical statements. Consider using 'substantial'.

Thank you. This has also been implemented in other places as well.

L34 'and are known'

This sentence has been clarified.

L35 I believe biomass burning is also emitted at the surface and you mean it is moved into the upper atmosphere.

I would argue that the emission also does not occur at the surface, but instead occurs at wherever the material being combusted is in direct contact with the atmosphere, whether it is bubbles formed under the soil at the intersection of oxygen and peat, or it is in pieces of lofted grass which not yet fully burned but are caught in the uprising atmospheric plume and finally combust far above the surface.

The point that we all agree on is clear however: the emissions occur into parcels of air which rise at a sufficiently rapid rate that they are for all effective purposes of the measurements employed in this work (MISR, OMI, MOPITT, MERRA, and MODIS), "emitted" into the atmosphere at a given height. Sentence 35 has now been edited to reflect this.

L40 The statement that aerosols above the PBL have a bigger influence on the atmo- sphere may be true in some context, and the authors do provide citations, but they need to be a bit more specific here. I assume they mean in some sort of normal-ized sense (eg Pinatubo had a big influence on global mean temperature, but in an integrated sense aerosol in the boundary layer probably has a bigger impact). Either way, while a very interesting point to make, the authors might want to expand on this statement a bit for clarity.

This is the issue of radiative forcing. In this paper, we are looking at remotely sensed measurements on a scale of 1km to 100km, and hence at the implied radiative forcings at these scales. A very interesting topic for another time. The review paper included Tao et al. 2012 (already cited) is an excellent introduction to this topic.

L45 Who used? I think the authors have a typo and all the citations have stuck together.

Thank you.

L53 Lidar isn't capitalized: https://www-calipso.larc.nasa.gov/

Thank you for this correction. This has been implemented in 2 places.

L81 Large majority is redundant

Updated.

L99 typo, remove 'the'

Thank you.

L144 Specifically

Thank you.

L145 Does this mean that when you have cloud or aerosol you don't get CO measure- ments?

This is now clarified in detail based on a question from Reviewer #2.

L156 NO2 also has substantial industrial sources. The way that this is written implies that NO2 is only from fires.

We did not mean to imply that $NO_2$ does not have a significant urban source. We fully agree that $NO_2$ has a significant urban source. But we stated that the temporal-spatial distribution of urban NO2 is much lower than for fires, because other than transportation sources, most urban sources occur in fixed locations, and even transportation sources tend to follow fixed pathways (roads, shipping lines, air routes, etc.). This has been clarified.

L187 Note that inputs are not necessarily orthogonal, unless you pretreat inputs some- how. For example, NO2/CO is going to be correlated with NO2 and CO.

This has been explained above.

L188 Typo in this sentence.

Corrected.

L216 This sentence is very unclear- how are you 'injecting additional information'? As you say earlier all data sets have to be present. This seems to imply that data points with missing data will sometimes be considered and additional information will sometimes be 'injected'.

This sentence has been changed and broken into two.

L218 It is also unclear how you intend to reduce bias. Do you mean that you will try out data sets that measure the same quantity to get an estimate of bias.

See above.

L254 It would be good to define FRP somewhere in the intro or methods in terms of its physics (for people outside the biomass burning community).

The following has been added into section 2.7
FRP is the measure of the radiative energy released by the fire. It is usually found in the infrared part of the spectrum as this is the part of the EM spectrum that corresponds closely with the temperatures that fires occur at in the Earth System.

L270 something that I think needs to be discussed in the use of this plume rise model is that it is based on a model from 1965. In the methods there need to be a few sentences on why this model has not been improved upon since then, or why it is an appropriate comparison to the MLR model. Not discussing this runs the risk of making the plume model seem like a straw man to those outside the plume modelling community. Another aspect of this plume rise model is that earlier the authors state that it begins to fail for small fires. The analysis should really be subset to fires that satisfy the assumptions going into the model, rather than degrading the model with fires that the plume rise model is not designed for.

First off, the reason why the plume model fails for small fires is not because of an inherent problem with the plume rise model itself. It is with the fact that small fires are frequently missed altogether, or have their FRPs severely underestimated. This is not a problem with the plume rise model itself, but of the inputs being used inside of the plume rise model. Another issue is the resolution at which MERRA and most reanalysis meteorological products release their temperature and wind profiles, leading to too coarse of a resolution. You are right that a deeper analysis may be helpful. However, this was done in a previous paper we authored (Cohen et al., 2018) and we are not sure if copying and pasting that would be helpful here or not.

However, to more fully address this issue, many such corrections and additions have been made throughout the text, as outlined both above and below. Please let me know if you think that these changes are sufficient.

L329 A citation to a review article here might be helpful.

A review of this has been added, Gunturu et al., 2009.

L332 Different than each other? Do you mean when the plume model and the mea- surements? If this is the case this also seems fairly arbitrary to be testing the model and throwing out the results when they are poor.

All of the data for the plume rise model is now included, whether the region fits well or not. Therefore, this sentence is removed.

L340 Is this just a function of bias from the plume rise model treating fires that are smaller and thus don't satisfy assumptions in the model?

This is not true. There is no bias in terms of the plume rise model being able to handle smaller fires, the problem is that smaller fires tend to have their FRP and other remotely sensed characteristics biased, since

485 they are too small as compared to the spatial and temporal assumptions underlying the fields being measured.

L341 how well the data what?

490 Multiple changes have been made to these paragraphs. The finding is that it is the higher rising fires which are not reproduced by the Plume Rise Model, which is the exact opposite of what the reviewer and the community have focused on in the past. Again, this supports the conclusions made here that it is in fact missing physical forces, some extreme form of underestimation of FRP for medium and large fires, or a combination of these factors that is driving these differences.

495 The following is the partially retained and partially edited paragraph in Section 3.2

Next, we look at the difference from day-to-day at each of the sites which has a mean value less than or equal to 0.25 km. Using these results, we find that the mean daily difference between the plume rise model and the MISR measurements as a whole show a large amount of variation, with a global average
500 of 0.44 km, a maximum of 1.13 km (in West Siberia), and a minimum of 0.04 km (in Argentina). Across all of the different regions we find that the plume rise model underestimates the plume height. Furthermore, we find that the differences between the Plum Rise Model and MISR are not normally distributed, with higher values not being able to be reproduced under any conditions, strongly indicative of a bias, in that somehow the largest, hottest, or most radiatively active fires are those being not
505 reproduced well by the Plume Rise Model. In addition to this, we compute the RMS error (Table 3) as a way of quantifying overall how well the model and MISR match. The RMS is found to be considerably larger than the difference of the means, indicating that a small number of extreme values are dominating the overall results, which were found to be 0.67 km, 0.88 km, 1.36 km, 0.40 km, and 0.85 km in the respective five areas.

510

L344 While I understand the attraction of minimizing the number of figures, but this article only has 3 in the main text. I feel that the PDFs of modeled and observed plume heights could be moved to the main text.

515 The PDFs of the observed plume heights, along with many other figures, have been moved into the main text. All of the underlying data, including plots in the supplemental information, are available at:

As included in the Code/Data availability statement:
https://doi.org/10.6084/m9.figshare.10252526.v1 and https://doi.org/10.6084/m9.figshare.12386135.v1
520

L365 How does the analysis account for times when the area is very crowded with burning? How does it tell where plumes actually originate from? Can a plume from another fire be mistagged or affect plumes from a nearby fire?
525

This is a fair point. We have relied on the MISR data as being able to distinguish the individual plumes. However, a fair argument was made by Cohen et al., (2018) that this question is actually not the right one to ask. In reality, if an instrument such as MISR cannot distinguish the plumes from each other, then effectively, as far as any modeling system will be able to capture, or the atmosphere will be able to feel,
530 they are a single plume. This has been discussed at length in the paper cited above. The only case in which this would possibly matter is if there is a bias between the plume height at equilibrium locally and that of

a plume cloud regionally. However, one could argue that if the fires are packed so tightly, that they should be measured as a group and not individually.

L375 A clear list of assumptions in the methods would be good. I as- sume there is more than one plume rise model in the literature (for example https://link.springer.com/article/10.1007/s10661-005-1611-y). The authors must show results from at least two leading plume rise models to show that the poor results of the 1965 model are not just due to poor construction of the model and limitations in what it can do (and applying the model outside of its assumed conditions).

We have read this interesting review article carefully and have found that it supports our conclusion. In fact, even the plume rise model we are employing was not discussed. In fact, the only ways they have discussed are using mesoscale models (similar to the vertical approach employed by Cohen and Prinn, 2011), global scale models (similar to the vertical approach employed by Cohen and Wang, 2014), and reanalysis products (similar to MERRA as employed here). We have included in results of measurement constrained studies using lidar as well, and found that such methods still fail, in that they are training models of the same type.

We have included the following sentence in section 1
    Large-scale reviews of the biomass burning literature spend a lot of time on how the atmosphere impacts the burning conditions, but also tend to overlook the issue of how the emissions are rapidly vertically distributed upon being emitted (Palacios-Orueta, et al. 2005).

L385 Is this because Argentina is dominated by the Pampas and fires tend to be over large areas and are uniform and the meteorology is relatively less complex?

Yes, this is also consistent with the results as demonstrated by Table 4, Figure 6, and Supplemental Figure S6.

We have edited and expanded upon the text to include the following sentence in Section 3.2
    It is under these relatively lesser polluted conditions, where the fires are fewer and/or less intense, where a lower amount of total material is being burned on a per day basis of time over the total surface area burning, or where the meteorology and the vertical thermodynamic structure of the atmosphere are more uniform, that the plume rise model can achieve its best results (Table 4, Fig 6 and Fig S6), and thus that the plume rise model is reasonable to use in such a region.

[Figure]

[Figure]

[Figure]

**Figure 6: PDFs of the NCEP reanalysis vertical temperature gradient d[K]/d[km] over the locations and days that contain MISR plumes. The 8 regions over which the regression model is valid are shown.**

L397 I think rather than coming up with 7 combinations of predictors a better approach might be to only have one model with all the predictors or use some sort of objective algorithm (eg machine learning) to remove low explained variance predictors. Arbitrar- ily coming up with 7 models seems like it will almost always guarantee a model works well.

Answered above.

L408 Fragment

Corrected.

L411 Again, I don't understand how this is an evaluation if predictions that agree too poorly are removed.

Explained above.

L430 The three regions shown in Fig 3 are for a few plumes (judging by plotted data points) and for only

a subset of the plumes in Fig1.

Figure 3 has been expanded.

595

L478 Which of the regression models is the new method?

Regression models 1 through 6 are new. The most useful are always regression models 1, 2, or 4. This has been explained in much more detail above.

600

L483 What are the 'modelled results' in contrast to the plume and regression models?

This has already been changed elsewhere.

605 L497 Somewhere there needs to a scatter plot of MLR model plume height versus observations. One possibility is that you are just fitting the mean. The MLR model is guaranteed to do this well (it minimizes unexplained variance). To do this correctly you should train the model on one region and apply it to other regions to get rid of the overfitting problem.

610 This has been explained above, and the results can be found in Figure 3, Figure 4, Figure 5, Figure 6, Table 4.

Fig1 I am not sure how useful this plot is because the dots obscure the land surface type.

615 We have updated Figure 1 with this

[Figure]

**Figure 1: Land surface type at each of the daily MISR measurements from January 2008 to June 2011. Each dot corresponds to an individual aerosol plume, with different colors representing different years.**

620

Fig2 Please use some different line styles and markers. Most of these colors are indistinguishable.

An excellent comment. We have made some changes here.

625    The following are now used for Figure 2

[Figure]

[Figure]

(d)

630    **Figure 2: PDFs of all daily MISR plume height measurements from January 2008 through June 2011 (which are 5000m or less) over each of the following geographic regions: (a) Africa, (b) Eurasian High Latitudes, (c) Tropical Asia, and (d) the Americas. Solid lines correspond to regions which have a successful regression model, while dashed lines are regions which do not.**

Response to Author 2:

I will keep this short and to the point. I think the basic idea of trying to investigate the relationship between trace gas/aerosol plume height and the pollutant loading is good. But having read the manuscript few times, I do not believe the authors have approached the problem with the right tools. My opinion/review is mostly from the observational perspective and I don't know much about the plume models.

1) Why use the total column values of NO2 and CO, when the authors themselves C1 show how, depending on the region, aerosols can be lifted to different heights. What do we actually scientifically gain by looking at the total column only? It is not a surprise that when episodes of strong pollution occur (e.g. fires, biomass burning), the total column values will increase and depending on the thermodynamical conditions (e.g. strength of convection) the lofting will occur. I understand that the vertically resolved observations of NO2 are not available, but altitude-resolved CO retrievals are available from a number of sensors, MOPITT, AIRS, IASI etc. I also wonder why the authors don't use aerosol layer heights from CALIPSO (possibly combined with OMI)? Wouldn't that be the most accurate account of plume heights?

We have introduced the results from MERRA into the paper, based on comments from the first reviewer. We do agree that additional vertical measurements from MOPITT would be interesting to investigate as well, but instead propose this for a future effort. One reason for this is that the horizontal and vertical resolution of MOPITT are very challenging to use unless very carefully applied, which would go beyond the current time allotted for this major revision. Furthermore, we are using actual measurements of height from MISR, and first wanted to see if the simpler column loadings would be representative. This further is consistent with recent findings from my team as just published in Lin et al., 2020a. which have shown that the column loading of highly variable regions of CO map very well with biomass burning events, as well as Lin et al., (Under Revision) 2020b; and Cohen et al., 2018, in which we have further looked into the MOPITT vertical distribution associated with global biomass burning, although at temporal scales of a week to months, not day-to-day as we are working on here. The suggestion of using CALIOP is also very interesting, and we believe that based on the results from Cohen et al., 2018, it would yield significant findings. Again, we did not have enough time in this current revision round to accomplish this, especially since finding a sufficiently large number of overpasses in a region which is actually influenced by the plumes, not merely over an "average region" is incredibly challenging work. However, we appreciate this suggestion and will seriously look forward in the future to address this.

As per your suggestion, we have carefully checked the NCEP vertical temperature gradient as a proxy of the thermodynamic conditions (e.g. strength of convection) and the vertical air mass rise at the surface.

Based on these findings, we have added the following to the paper in Section 3.5 and Figure 6
In terms of the magnitudes of the vertical temperature gradient (dT/dz) and the vertical wind speed at the surface, we have not found any correlation or relationship between the cases in which the regression model performs better or worse. Even considering those cases in which there are extremely atypical values in these variables, such as positive temperature gradients (i.e. an unstable atmosphere), or negative temperature gradients which are more negative than the -9.8 K/km rate which is the pure dry air thermodynamic limit (i.e. extreme stabilization due to intense aerosol/cloud cooling), as observed in Fig. 6. This provides a further piece of support to the idea that the regression model works well under conditions where there is some local non-linear forcing in the system which is not being taken into account,

whether it is a coupled chemical, aerosol dynamical/size, radiative-dynamic, thermodynamic, or direct/semi-direct/indirect type of aerosol effect, all of which are being accounted for to some degree by the loadings of $NO_2$ and CO, but which are missed by the model underlying the meteorological reanalysis data (e.g. Cohen et al., 2011; Wang et al., 2009).

However, it does seem that under the conditions where the regression model was not able to be formed, that there are some important differences in terms specifically of the vertical temperature gradient variable. In specific, in the cases in which the value of dT/dz is either more negative than -9 K/km or positive, that the MERRA results are far better than those from the plume rise model, as compared to not under those conditions. However, such cases only account for 15% or fewer of the total cases observed in this study, and therefore do not play an outsized role.

2) The lifetimes of CO and NO2 are very different. CO has much more homogenized distribution in the atmosphere, especially as the altitude increases due to transport processes etc. So can the authors disentangle this background signal from the one that is associated with the biomass burning plumes for CO, especially over those regions that already have strong background variability in industrial+traffic pollution?

This is a great point and one of the reasons why we also wanted to choose to use both $NO_2$ and CO simultaneously.

We have included the following text in section 3.3, Figure 5, and Table 1

Due to the fact that NO2 and CO have very different lifetimes in the atmosphere, a fire-based source is expected to have a high level of both CO and NO2 close to its source, which decays as one heads away in space from the source. This decay should be a function of the wind direction as well, as both the CO and NO2 upwind will not have a significant source, but downwind the CO will have a significant source, as shown in Fig. 5. We find that our results are consistent with this theory. First, we have found that the regions that have the highest $NO_2$ at the same time as the MISR measurements are made, also have a very strong overlap well with the locations of the MISR plume heights. We further determine this to be true for every year on a year-by-year basis (Fig S1). Second, we find that the higher values of CO match well with the year-to-year locations of MISR fires (or downwind thereof) at most of the sites, including in Alaska, Central Canada, Central Siberia, East Europe, East Siberia, Northern Southeast Asia, Siberia and North China, and South America. As expected, there greater smearing away from the source regions. As expected, this is due to the fact that the lifetime of CO is much greater.

Furthermore, in terms of changes in time, a climatology of CO should be slightly higher due to the added emissions from the fires, but the NO2 should be much larger than the climatology, since there is little to no retention in the air, as demonstrated in Table 1. To account for this, we have also looked at the difference between the fire times and the long-term climatology. Over regions which are urban and hence contributing randomly to the variance, we expect the differences to be smaller than due to the fires, and this is observed clearly as well. These results are also shown to be consistent with recent work (Cohen, 2014; Lin et al., 2014; Lin et al., 2020a), showing that the characteristics of the spatial-temporal variability of fires is quite different from that of urban areas, and has a much higher variability both week-to-week and inter-annually.

Thirdly, this is pointed out in the time series plots **(Figure S1)**, where the CO and $NO_2$ are both considerably higher during the fire times than the rest of the year, while at the same time, the $NO_2$ and CO are both higher over the subset of points that have fires on the fire days than over the entire region on the fire only days. The idea of a proper study to disentangle the downwind regions from fires, downwind

regions contaminated by both urban regions and fires, and downwind regions from urban-only regions is
something of merit and would be an excellent follow-up work. This part of the response will not go into
the main paper.

The following is now included as Figure 5

[Figure]

[Figure]

**Figure 5: Spatial distribution of annual compilation of all MISR fires (magenta dots), mean OMI NO₂ column loading on days where there are fires (black isopleths [\*10¹⁵ mol/cm²]), and mean MOPITT CO column loading on days where there are fires (Colorbar, mol/cm²). The corresponding regions are: (a) 2010 Central Canada, (b) 2010 East Europe, (c) 2009 and (d) 2010 Northern Southeast Asia, (e) 2010 East Siberia, (f) 2008 and (g) 2011 Siberia and Northern China, and (h) 2010 South America.**

3) There is virtually no description of how different satellite data products are quality controlled, analysed etc. The devil is in the details. What quality flags are used? How are cloudy/non-cloudy cases handled? Is there a consistency in such cases across all datasets? How is the sampling affected by the quality control?

To add in more details, the following have been added at the respective parts of the manuscript in sections 2.3, 2.4, and 2.7. Additional corrections have been made in 2.3 to reflect the updated version of the CO data used.

The following has been added to section 2.3
In terms of the CO from MOPITT, we take the day time only retrievals (to reduce bias) from version 8, level 3 data. In specific we use the combined thermal and near infrared product (Deeter et al, 2017). We further constrain the data to where the cloud fraction is less than 0.3 and where the vertical degrees of freedom are larger than 1.5. This combination has been shown to allow us to trust that there is a sufficient amount of signal and knowledge to demonstrate an actual measurement in the vertical, as compared with a result only dependent on the a priori model, as shown in Lin et al. (2020a). There are further gaps in the data due to orbital locations and very high aerosol conditions, all of which prevent entire coverage of our areas of interest each day. Therefore, we average all of the individual MOPITT data that passes our test to a 1°x1° grid.

The following has been added to section 2.4
In terms of the NO₂ from OMI, we first take the daily retrievals under the conditions where the cloud fraction is less than 0.3. Next, we aggregate the data to 0.25° x 0.25° using a linear interpolation and area weighted approach. In this way, those measurements near the edge of the swath or which are adjacent to cloudy areas are weighted less heavily in terms of the merged product. However, the areas are sufficiently large as to be roughly representative of the emissions from biomass burning of the NO₂ from within the grid box, as compared to that transferred from adjacent grid boxes.

Furthermore, for our computations we only retain those measurements in which we have data at the place of interest from MOPITT, OMI, and MISR at the same time. If just one of the three measurement platforms is more than 30% cloud covered, is not able to measure due to extremely high AOD levels, or is found outside of the swath at the given time, then that day's data is discarded in terms of developing

and the regression model, and any subsequent analysis. However, we do use all available data every day from within the respective boxes in terms of understanding the background values, and trying to better constrain the differences between the values of the column measurements over the identified biomass burning points based on MISR and those which are within the same larger area but are upwind, downwind, or not involved with burning at all. This is completely consistent with the fact that biomass burning is sub-grid within each individual respective 1º x 1º box for CO and 0.25º x 0.25º box for NO2, while simultaneously only occurring over a distinct of set of days.

[revised manuscript text omitted]

---

## Author Response (AR2)

Dear Editor, Reviewer 1, Reviewer 2 and Reviewer 3,

Thank you all for taking so much time to provide your meaningful and insightful comments and suggestions. We have taken them all into serious consideration and have strived to work hard to address them point-by-point in this response, your original comments are given (unedited) in yellow highlight, our responses are given in blue highlight and updates to the paper *are given in italics*. Thank you again for your time and deep insights! With your support and nudging, we believe that the results will make both a meaningful and important contribution to the academic discourse. Many thanks again!

Best Regards,

Jason Cohen

ACP, July 2020
Review of Wang, Cohen, Lin, and Deng,

Constraining the relationships between aerosol height, aerosol optical depth, and total column trace gas measurements using remote sensing and models

Overall, this paper includes a novel idea of incorporating CO and NO2 column-concentration measurements, as well as CO/NO2, in a statistical regression approach to modeling plume rise. They input MOPITT CO, and NO2 from the OMI instrument, and use MISR aerosol data for model validation.

Thank you very much. This was our intention from the start with this paper, and you have summarized this point very well. If we need to make any bullet points about its importance or impact, we will use your wording, if it is OK with you.

However, the paper needs work, in my opinion. There are a few overall considerations that might warrant further attention, and some smaller technical issues; key points are given below.

I believe it is important to point out that this review is based on the originally submitted paper, not the highly edited version which the initial two reviewers re-reviewed. For this reason, it has taken a bit more time to carefully address the changes, since some were already previously changed. However, due to the important recommendations provided, I am trying my best to address these points in full. Please excuse anything which may have been missed.

**Top-Level Notes**:

Greater care is needed in separating background aerosol, including aged smoke and pollution, from smoke emanating from specific fires. Note that "pollution" often indicates anthropogenic origin, whereas the term appears to be used here to include aged (and possibly also some fresh) fire emissions, as well as background emissions from other sources. It would be helpful to make clear what definition

is being used, and to make a distinction between fresh emissions (the target of the modeling here) and background emissions, either from earlier fires or from other sources.

We have analyzed both the loading of the $NO_2$ column measurements and the ratio of the $NO_2/CO$ column measurements over both those sites which only have measurements available of FRP and overall over the larger regions (Table S2). From these results, we find a few pieces of information. First, that the results are very similar in terms of the $NO_2/CO$ column ratios over both those specific points containing measured FRP values, and those points within the fire plumes as a whole. This is an indication that there is not a significant amount of urban emissions of $NO_2$ found in any of these regions at least from a statistically relevant point of view. If there were, the $NO_2/CO$ ratio would rapidly diverge between these approaches, due to the much faster chemical titration of $NO_2$ compared to CO. Second, we find that in general the ratio of $NO_2/CO$ falls into three separate ranges (defined by the central 80% of the pdf). The first group (Siberia and North China & Central Canada) range from $1x10^{-4}$ to $9x10^{-4}$, the second group ranges from $2x10^{-4}$ to $15 x10^{-4}$-$20x10^{-4}$, and the third group (South America) ranges from $6 x10^{-4}$ to $43x10^{-4}$. This is a strong indication that the ratio of $NO_2/CO$ is representing a physical meaning, possibly connected with the dryness and flaming fraction, the temperature of burning, the type of biomass being burned, etc. Furthermore, we find that separating the data into the high and low $NO_2/CO$ ratio values and then comparing the PDFs of the MISR height distribution and modeled height distributions leads to a clear improvement of the regression model as compared to the plume rise model (Figure S7). We have added the following to the paper in three different locations.

55

60

*On top of this, the $NO_2$ column loading and the ratio of the $NO_2/CO$ column measurements over only the selected grids which have available of FRP measurements, and over the larger regions as given in Figure 5 are found to generally be consistent, with the ratio found to be more so (Table S2). This indicates the $NO_2/CO$ column ratio over the fire regions tends to be consistent with the fire plumes as a whole, and is not found to be significantly influenced by urban sources of $NO_2$, which would lead to a vastly faster chemical titration of $NO_2$ compared to CO.*

65

*Following these ideas, the idea of characterizing the by the ratio of $NO_2/CO$ is found to nicely separate the data into three different groups, based on the bands generated by the central 80% of each respective region's $NO_2/CO$ pdf. Group 1 consisting of Siberia and North China & Central Canada, has a $NO_2/CO$ range from $1x10^{-4}$ to $9x10^{-4}$. Group 2 consisting of the remaining regions, has a $NO_2/CO$ range from $2x10^{-4}$ to $15 x10^{-4}$ to $20x10^{-4}$. Group 3 consisting of South America, has a $NO_2/CO$ range from $6 x10^{-4}$ to $43x10^{-4}$. This strong differentiation is consistent with the ratio of $NO2/CO$ representing a physical meaning, but being a single, continuous variable connected with the temperature of the burning, the wetness of the burning material, the latent heat flux, and the type and amount of biomass being burned.*

70

75

 *... "under all conditions, and even more so at the respective top and bottom 10% of each respective range of NO$_2$/CO ratio, in which the subset of regression model heights performs much better than the respective plume rise model heights when compared with the MISR height distribution (Figure S7)"*

Contributing to the issue of distinguishing sources, burning conditions often produce many small fires, possibly in addition to a few large ones. These might be difficult to represent accurately in a model.

This point is well taken. In the data we have used, we only have access to a single FRP value to represent the state of the total amount of fire within each pixel. Whether this single FRP value is due to many small fires found within that pixel, or a single large fire, we cannot distinguish. However, the vertical and horizontal meteorological measurements, such as wind and temperature are also derived from larger-scale average values, frequently 10s to 100s of km in size. The column measurements of OMI and MOPITT are also coarser than the FRP measurements. There is a good reason for this, because in the real atmosphere, small scale turbulence and mixing will impact smoke emitted from the fires, and cause them to merge. These assumptions of scale are built into the plume rise models themselves are based on the thermodynamics of the system, as compared to the eddy-scale dynamics. For these reasons, the overall results are not expected to be any better using higher resolution measurements unless they are guaranteed to be accurate, and unless an eddy resolving model is used. However, presently there are no remotely sensed measurements available to match the required spatial and temporal scales. Therefore, we have assumed that the data given within each pixel is distributed into a single fire (in the case of FRP) or is evenly distributed (in the case of wind speed and direction, temperature, and remotely sensed measurements).

Addressing this point, we have decided to look at the measured plume rise heights and model values associated with the top 5% of measured FRP values (FRP>132W/m2) as well as the bottom 5% of measured FRP values (FRP<8.5W/m2). Overall, we do observe that both the measured and modeled mean plume rise is slightly higher for the very high FRP values than in the case of the very low FRP values, as expected. Furthermore, we notice that there are more plumes found above the boundary layer in the measurements corresponding to very high FRP values than in the case of very low FRP values, although there still are plumes over the boundary layer in both cases. This is where there is a considerable amount of value added from using the regression models that include the remotely sensed measurements of NO$_2$ and CO, as compared to the plume rise model.

The following have been added to the paper:

*Analyzing the extremes of the FRP, leads to a top 5% of measured FRP of 132W/m$^2$ and a bottom 5% of measured FRP of 8.5W/m$^2$.*

*The high end of the FRP range of the observations in this work is not considered to be very hot in terms of fires, which should in theory help to reduce the known plume rise model bias of underpredicting very*

*strong FRP conditions, leading to an overall improvement in the plume rise model as analyzed in this work, as compared to when it is less constrained. As expected, there are more plumes found above the boundary layer in the measurements corresponding to very high FRP values than in the case of very low FRP values, although more importantly, there still are plumes found over the boundary layer in both cases, which is not expected based solely on the plume rise model formulation.*

Also, there are limitations in the degree to which the remote sensing instruments can resolve structure in the boundary layer (MISR plume-height uncertainty is between 250 and 500 m), and models are notoriously poor in simulating boundary layer height and vertical mixing timescales. So there are plenty of possible reasons why discrepancies between the "observed" and "modeled" boundary layer results might occur.

This is an important point to make. We have carefully re-analyzed the data (as presented in Figure S2) and determined that other than in Alaska, the differences between the MISR measurements, the plume rise measurements, and the regression model measurements in the boundary layer are always more than the 250m to 500m uncertainty range of the MISR measurements, meaning that the results as given in the other regions are consistent with the conclusions and assertions made. This said, your point about the uncertainty of the MISR measurements in the boundary layer is extremely important, and we have added more details to the text to address these measurement uncertainties that exist within the boundary layer.

*The only region over which the finding may not be statistically relevant is in Alaska, where the difference between regression measurements are all constrained to within a 500m height band, which falls into the MISR measurement uncertainty measurement range.*

Many 1-D plume rise models tend to overestimate injection heights for low plumes and underestimate them for higher plumes, thus diminishing the range of simulated height variation.

This observation of overestimating low FRP plume rise heights and underestimating high FRP plume rise heights was found in our own previous work as well (Cohen et al., 2018), and was one of the motivating reasons behind this present work.

As can be observed for our data (Table below), it seems that there basically are no conditions that exist for very high energy plumes in the first place (5% level is $8.5W/m^2$, while the 95% level is $132W/m^2$). This hence should negate the plume rise model's known tendency to underestimate higher plumes.

We believe that this point has already been addressed by the previous changes to the manuscript found above.

With a closer analysis of the reasons CO and NO2 column concentrations improve the correlation with
55 plume height in regression models, the current paper might shed some specific light on this issue.

It appears that for the third group (the highest $NO_2/CO$ ratio group) that the regression model works much better than the plume rise model, even at the extremes of the distribution. In some of the cases for group 2 (the moderate $NO_2/CO$ ratio group) that the regression model works considerably better than
60 the plume rise model, even at the extremes of the distribution. This indicates that under conditions that produce more $NO_2$ (i.e. hotter conditions) or that produce less CO (i.e. more oxygen deprived conditions) that the model works better in general (i.e. the overall mismatch with MISR is reduced). Overall the improvement is that the low biases are reduced, while the ability to reproduce a larger fraction of the plumes over the boundary layer is also increased.

65

We have already made changes to the text to address this issue.

**Detailed Notes**:

1. The references used in the **introduction** are a bit off. Although they are relevant to the subject of
70 plume-height retrieval, many are cited in the wrong context. A few cases are listed here. The authors might go over all the cited references and assure they are listed in the appropriate context;

Thank you very much for your recommendations and deeper list of references. We have made the changes recommended and extended upon those to other similar parts of the work. The following details
75 a point-by-point response.

1. For example, on Line 27, Val Martin et al. 2012 is about testing a plume-rise model using MISR plume heights, and Nelson et al. 2013 is about the MISR plume height algorithm. Yet they are referenced regarding "the impacts that aerosols have on clouds, radiation, the atmospheric
80 energy balance, and climate, human health, and ecosystems..." Although plume-rise does bear upon all these, the papers referenced do not cover those aspects.

   Addressed. These references have been moved to a more appropriate spot and different references have been added in here.
85

2. Mims et al. 2010 (Line 29) is about plume-rise in Australia, not about the atmospheric energy budget. An aerosol transport model result would probably be a more appropriate reference than Winker et al. 2013 (Line 31) when talking about the atmospheric distribution of aerosols in general.
90

   Addressed. These references have been moved to a more appropriate spot and different

references have been added in here.

3. On Line 37, Vernon et al. (2018) should probably be Flower and Kahn (2017; doi: /10.1016/j.jvolgeores.2017.03.010). The Alaska plume-rise results (Lines 47-48) are in Kahn et al. 2008, not Kahn et al. 2007. Val Martin et al. 2018 (Line 48) is about the MISR plume-height climatology, and although it mentions CALIPSO, that is not the point of this particular paper. (Maybe the Winker et al. 2013 reference belongs there.)

Addressed. These references have been added, updated, and/or moved to a more appropriate spot, while suggested references have been added in here.

4. On Lines 51-52, the climatology of plume heights in Val Martin et al. 2018 shows that smoke is injected above the boundary layer frequently at high latitudes in summer, and in a few other geographic locations, but in areas where agricultural burning is prevalent, high injection is rare.

This point is clearly made and clarified.

5. On Line 58, Ichoku and Ellison 2014 actually derive emission factors to estimate smoke source strength; they do not assess plume-rise models – that might be Val Martin et al., 2012 and Paugam 2016.

Thank you very much again. Your deep insights have helped us to better give credit to where it is due, and helped us to organize the referencing for those who ultimately read this work.

6. And it continues... it is up to the authors to go through all the references.

We have continued to make additional changes, as marked in the paper itself.

2. Lines 74-75. "The vast majority of biomass burning is a man-made activity..." This is true only in certain regions, not in general, and not globally.

We completely agree with the point about lightning induced fires, and have taken this into account in our edited version.

*The vast majority of biomass burning in the tropics and non-tropical agricultural regions of the world is a man-made activity (Kauffman et al., 2003; Achtemeier et al., 2011; Paugam et al., 2016), while in certain arctic regions, lighting accounts for a significant amount of biomass burning (Generoso et al., 2007).*

3. Lines 85-90. The primary reasons plume rise models have difficulty reproducing the actual aerosol vertical distribution in the atmosphere is difficulty in quantifying the heat flux and in modeling entrainment (Kahn et al., 2007, Val Martin et al., 2012). Removal mechanisms in general, and rainfall in particular, are relevant to longer distance transport and deposition, but

not to plume rise in most cases. Aerosol processing in polluted conditions is important for aerosol chemical evolution but is likely to be a dominant factor in plume rise modeling only in very special cases. This entire paragraph discusses factors relevant to transport and deposition, without making clear that it is no longer discussing plume rise modeling.

We believe that one of the problems with the current approach to plume rise modeling is that it is not considering the anthropogenic heat profile and emissions correctly. For more optically thick, the added heat from BC absorption is more significant than previously considered, especially so in the tropics. In highly humid air, the heat due to condensation of water vapor into an associated aerosol water phase is large and also not considered by the plume rise models. We do agree that for relatively thinner plumes, in relatively drier atmospheric conditions, with generally larger heat loadings at the surface that your point is well taken, even though as we have pointed out here, within the central 90% of the data, we do not have any extremely high measurements of FRP. We just want to be fair about the fact that there are important aspects of the thermodynamics that are related to plume rise modeling, which have not been considered in traditional plume rise modeling. To make this point clearer, we have added the following sentence, following the reviewer's recommendation:

*The present generation of models has difficulty to reproduce the actual vertical distribution of atmospheric aerosols when addressing cases that do not tend to have a combination of a highly energetic fire source, a relatively dry atmosphere, and a relatively optically thin smoke column emitted by the fire.*

4.  Lines 94-95. Again, the references don't seem to address the context. And it is not clear what type of simple models are being discussed (plume rise or transport and deposition).

We have inserted the words *plume rise* to make this clear.

5.  Another paper that might be worth reviewing in the context of this study is Heald et al. 2004, doi:10.1029/2004JD005185. And the following might also be of help: Kahn 2020 doi.org/10.1029/2020EO138260.

Thank you for supplying these excellent works. The last sentence of the paragraph has been expanded, and now reads as follows:

*In summary these factors can lead to actual changes in the vertical distribution of aerosols that simple models are not able to reproduce, including those which have used inverse modeling with a fixed vertical a priori (i.e. Heald et al. 2004; Cohen and Prinn, 2011), in turn affecting the distribution of aerosols hundreds to thousands of kilometers downwind, supporting new measurement-based perspectives (i.e. Kahn 2020).*

6.  Lines 129-133. Given the diversity of burning conditions in boreal forest, tropical forest, grassland, agricultural land, etc., it is not clear why one would look at global average variability rather than stratifying at least into these general categories.

Our findings seem to be that the $NO_2/CO$ ratio is less dependent on the different land use types than it is on other factors, some of which are still not clearly known. We argue that from the

atmospheric perspective, and the perspective of the ultimate vertical distribution of the aerosols from fires, that the $NO_2/CO$ ratio may be more important. It would be great to dig more deeply into these various driving forces in a future project. Thank you for the suggestion!

7. Lines 134-137. Rather than something deeper, the distribution of burning reported here seems to just reflect that the subtropics are dominated by un-vegetated desert.

Our work has found and analyzed regions in the subtropics including: Argentina, Southern Eastern Europe, and Northeastern China. However, it is true that urban emissions sources dominate in many of the subtropical regions, including in the USA, Central China, and Southern Europe. These regions may therefore be missed using the variance maximization approach, where the variation induced by fire emissions are masked due to random variations induced by the considerably larger urban sources. This insight just means that a new perspective will be required to adapt the finding here to regions which have a mixture of urban and fire sources, which for CO means very large areas, as its lifetime is relatively long.

8. Line 147. The reason given for averaging MOPITT to 1x1 degree does not seem to be justified by the statement that the data are affected by clouds and orbital limitations. MISR data are also affected by clouds, MISR is in the same orbit as MOPITT, and MOPITT actually has a wider swath, yet the MISR data are averaged to 10 x 10 km in this study.

CO is a much longer-lived species in-situ as compared to aerosols. Therefore, geospatial variance is much smaller due to chemistry and physics. Hence, such averaging has been shown to be reasonable. This may be a limiting rationale behind why it is hard to separate smaller fire sources from larger urban sources in the mid-latitudes. However, it has no impact on the results given in this work. Again, a very interesting thing to follow-up on with respect to the spatial and temporal resolution issues.

9. Line 158. Note that NO2 is produced by a range of anthropogenic activities, so as the focus here is on wildfires, one must be careful not to assign to wildfires NO2 produced by other sources.

Due to the higher spatial resolution of OMI and the much shorter lifetime of $NO_2$ as compared to CO, this is much easier to address. It is possible that there are some co-variations in the $NO_2$ emissions that correspond with the same temporal variation as the fires, but they are not found to make a significant impact in this work. However, the point is well regarded, and would lead overall to an underestimation of the spatial distribution of the overall fires discussed in this paper, with an example being the biomass burning of transported wood occurring in urban areas not be differentiable within this context. Perhaps an excellent follow-up work to look at in terms of upward looking AOD and $NO_2$ measurement platforms?

10. Lines 162-163. There are many other differences between wildfire emissions and those from urban combustion sources.

While there may be differences in the ratios of carbon, nitrogen, and oxygen between biomass and other urban combustion sources, these do not lead to such huge differences in the output emissions. The majority of the differences, with respect to CO and $NO_2$ emissions stem from the environment and/or way in which these sources are burnt. For example, biomass fuel cut from a

tree, dried out, turned to charcoal, and ultimately used for cooking may be quite similar to coal used for cooking, although one is biomass and the other is urban. The critical issue here is using the geospatial and temporal variations in the amounts and ratios of $NO_2$ and CO as measured, to attempt to understand the differences in the sources and the ultimate rise of the plume heights. A slight change will be made to the paper, so as to be slightly clearer with respect to the point about the environment in which the burning is occurring.

*Although emissions from biomass burning are similar to those from urban combustion sources, there are two major differences, arising from the much higher burning temperature and the environment in which the combustion occurred.*

11. Line 169-173. Note that the Freitas (2007) model was developed in part to address some of the limitations of the Briggs model. So was Sofiev et al. (2012, doi:10.5194/acp- 12-1995-2012), which probably should be contrasted with the approach given in Section 2.7 of the current paper.

This point is important. To address this issue, the last revision added an entire section on comparison against the MERRA reanalysis. This reanalysis uses a model which includes CAPE type of motions, and therefore would be similar in nature. This has already been explained in its entire own section, 2.8. A reference to Sofiev et al., 2012 is added.

12. Line 177. You might say more about the MODIS data used; "MODIS hot-spot information" is not enough.

This was addressed in the previous revision.

13. Lines 179-185. A key input to the plume rise model would be the dynamical heat flux, or some other measure of the buoyancy generated by the fire. As this would not come from the NCEP reanalysis, does this come in some way from the MODIS hot-spot information?

Yes, from the MODIS hot-spot information. This point was addressed in the previous revision.

14. Lines 223-224. Figure 2 is very difficult to interpret. The axes should be labeled, and the number of counts in each region should be given, maybe in the legend or in Table S1 (or both). Also, some smoothing or an expanded vertical axis might help in eliciting the main patterns from the mass of lines in this figure.

This change was made in the previous revision. It was divided into 4 separate plots and the axes were enhanced.

15. Line 225. An injection height of 29 km seems very unlikely. The most extreme PyroCb on record reached no more than 3-5 km above the tropopause, to about 17 km in that case. The extreme height seems more likely to be an error, perhaps in co-registration of the multi-angle images.

Point well taken. The following has been added:

*(with extremely high values in the middle stratosphere possibly an error),*

16. Line 231. If ~8% of the plumes in this study are above 5 km, and they are likely to be concentrated in certain regions, wouldn't that be important to include in the study, especially as these would have to be major events, possibly generating a disproportionate amount of smoke?

The newly revised lines, encompassing both the 2nd revision and your comments are given below. As you can see, the number of such very high plumes is very tiny.

*Due to the fact that first, the majority of the plumes are injected into the boundary layer or the lower free troposphere, second that this paper is not looking into the underlying physics of stratospheric injection (Pengfei Yu et al., 2019), and third that plumes tend to accumulate within layers of relative atmospheric stability, therefore an upper bound cutoff on the measured values of 5000m is imposed. This is consistent with the fact that over the regions of interest in this work, fewer than 0.48% of the total plume heights are more than 5 km.*

17. Line 240. The vertical resolution of the MISR plume height retrievals is between 250 and 500 m, so how does the data set report percentages of plumes below 200 m?

Thank you for raising this excellent point. Based on your knowledge of the data, we have adjusted these calculations to 500m, so as to ensure that plumes below the 24-hour boundary layer height are constrained based on both the daytime boundary layer height (which may be as low as 200m) and the measurement uncertainty. We have altered the text as below, including the new calculations:

*To safely consider those plumes which are definitively near the surface (i.e. never above the boundary layer) a plume height below 200m would roughly corresponds to the boundary layer maximum in the middle of the day (Guo et al., 2019). However, due to the measurement uncertainty of the MISR heights being between 250m and 500m, instead the percentage of total plumes with a height below 500m is chosen, which is found to have a total percentage of respective plume heights of: global (11%) and a range from a minimum of 0.68% in Southern Africa to 49% in Argentina. Given the diversity of these results, there is a need to more deeply understand the driving factors across all of these different regions, as well as the importance of biomass burning in terms of transporting aerosols through the boundary layer.*

18. Lines 244-253. Two limitations of the MISR data should be noted. First, as the MISR swath is about 380 km wide, locations on Earth are sampled on average about once per week, which still captures many fires, but misses many others. Second, the MISR overpass is at about 10:30 AM local time, whereas biomass burning tends to peak in the late afternoon. These limitations do not invalidate the work, but they should be taken into consideration when presenting global statistical conclusions. For example, the burning season might appear truncated if much of the actual burning occurs only later in the day than the MISR overpass.

We agree that pointing out clearly some of the limitations of the measurements only strengthens the results. However, as the co-author on a paper currently under review looking exactly at the time of day of overpass issue, I am not sure that I completely agree with your assessment that morning overpasses have a bias compared with afternoon overpasses, although in some regions

this seems to be the case. Regardless, we have added and/or changed the text in the following way, to reflect the limitations in the coverage from MISR.

*Furthermore, MISR has a relatively narrow swath, not providing daily coverage to all points, coupled with a morning overpass time which may lead to negative bias in some regions and positive bias in other regions in terms of observed fires. This combination allows us to clearly demonstrate that the observed smoke peaks are in fact due to burning of a significant amount of material, and are true cases of biomass burning, while not possibly being fully representative of all biomass burning events. In the observed cases,*

19. Lines 259-269. FRP is only loosely correlated with plume buoyancy, for specific physical reasons; the dominant ones are: (1) partly filled pixels; (2) non-zero smoke opacity above the fire front (which you mention), and (3) fire emissivity less than 1 (e.g., smoldering). This results in loose correlation between FRP and injection height. See, e.g., Kahn et al., 2007 or Val Martin et al. 2012, Figures 7 and 8 and associated text.

The point about partly filled pixels has been added and the references updated. The perspective taken by the plume rise models is that true fire radiative power should be strongly related to plume buoyancy at the surface, while these factors create a discrepancy between the true FRP and the measured FRP.

*On top of this, there may be partially filled pixels in the remotely sensed measurements (Kahn et al., 2007; Val Martin et al. 2012).*

20. Lines 283-284. Gas emissions are related not solely on the buoyancy generated. In addition to the factors mentioned, different ratios of CO and NO2, depend, for example, on fuel type and surface moisture.

This was already addressed from version 1 to version 2. Thank you!

21. Lines 294-300. Note that fire fronts are typically on much smaller scales (10s of meters) than the CO and NO2 measurements. In Northern China, high background CO and NO2 is likely due to anthropogenic activity, whereas in Siberia, it is likely from other fires. Often multiple fires of different sizes burn in an area where fuel load and fire weather conditions are favorable. You probably need to carefully separate any background contributions from those generated by the fire of interest to obtain a meaningful relationship between the gas concentrations and plume rise. Simply identifying above- background pixels and assuming the background contributions are negligible might not be good enough.

This important point was thoroughly addressed between version 1 and version 2. See figures 5 and S7.

22. Lines 323-328. Aerosol radiative effects, rainfall, etc., can be important for transported smoke, perhaps even a few hours after emission. But the buoyancy, ambient atmospheric structure, and entrainment are likely to dominate during plume injection.

This statement has been partially addressed between version 1 and version 2. Further support is

added to the text below.

*This finding is consistent with evidence that the vertical plume rise and distribution of tropical convective clouds is sometimes dominated by in-situ heating and turbulence even more so than the initial heat of condensation (Gunturu, 2010).*

23. Line 337. I've lost track of which plume rise model is being discussed here. I thought you were testing all seven regression models given in Equations 1-7.

In all cases, the plume rise model refers to the physical plume rise model, not Equations 1-7 (although Equation 7 is structurally similar to the plume rise model). The regression model, or Equation 1 to Equation 6 model (other than Equation 7) refers to the models based on the measurements of dynamical as well as remotely sensed measurements of $NO_2$ and/or CO.

24. Lines 352-355. Again, I'm unclear what model is being discussed. I do not recall discussion earlier of a plume rise model having a vertical resolution that would resolve better than 0.5 km in the boundary layer.

The plume rise model results frequently show a distribution of less than 500m, as demonstrated in Figure S2. This, combined with your information that the MISR measurements are uncertain from 250m to 500m is a motivating factor behind this work to better address the vertical height issue!

25. Lines 414-421. It is interesting that the column concentrations of CO and NO2 provide better correlation with plume height than FRP or wind. The CO and NO2 are not height- resolved, although in some cases these gases would be primarily generated by the fire in question, perhaps yielding an indication of fire intensity. It might be worthwhile to investigate this further, as it would offer a physical explanation for the observed correlations. Identifying the conditions when the relationship does not apply, perhaps cases where NO2 or CO from other sources dominates, for example, might also help in interpreting the data and improving the model.

This is a critical reason why we have used few of the locations in the extratropics. It is hard to find regions that we can be certain of, from a global analysis and perspective that are clearly dominantly influenced by biomass burning. It is hoped that this technique can be expanded by the community over specific regions using both high spatial resolution measurements from the new MISR products, TROPOMI, etc., or in addition with newer modeling efforts. We also look forward to attempting to further expand this work into areas which are more diverse in terms of their atmospheric background loadings.

26. Lines 435-437. Regarding the much smaller variance in the measured values, it might be worth remembering that the MISR plume heights are all acquired only around 10:30 AM local time.

The idea of a possible temporal mis-match was mentioned clearly before, as per your suggestion. It is quite interesting to note that MERRA does not provide the time information clearly, even though it is supposed to be assimilating the MISR data! We support your effort at clarity in this regard.

27. Lines 451-458. I think the reasons behind the performance of the regression model might be among the most important results of a study such as this one.

We appreciate this important point and have both edited existing text and added additional text to support this insight.

*This is further found to be true in the case where the data at the high end of the $NO_2/CO$ ratio profile are considered. This improvement is found in terms of both the bias and the RMS... These findings are consistent with real true world conditions, where there is a significant impact of co-emitted aerosols and/or heat, and these results with the $NO_2/CO$ ratio would hint that higher burning temperature conditions, or fewer oxygen limited conditions, may be important driving forces.*

28. Section 3.4. One way to assess regression model performance is by randomly selecting 10% or 20% of the data to leave out when the regression coefficients are calculated, and then comparing the model with against the withheld data.

We understand and appreciate this point. We also have found that the data is quite limited and may not be sufficient in some regions to begin with. Since this is a way to compare the regression model against its own fitting, and due to the lack of abundance of data, we feel that this exercise may be better suited to follow-up work analyzing the reasons for uncertainty behind these and other results. Thank you very much for this important point!

29. Lines 508-514. Accounting for the complex distribution of flaming and smoldering in many fire situations, especially larger ones, and modeling entrainment accordingly, are probably important if not dominant factors.

We feel that the reason why including the $NO_2/CO$ ratio specifically may be so important is that it can help us evaluate from a different perspective the issue of total biomass consumed (a function of CO) versus the atmospheric temperature at the point of combustion (a function of $NO_2$). Perhaps this new approach will help us to move beyond just flaming and smoldering, into a more diverse way to look at this issue. We have added the following:

*Thirdly, the finding that the $NO_2/CO$ ratio is extremely important in terms of matching the vertical distribution works to address the larger community issue of flaming versus smoldering in a more smooth and precise way, opening the possibility of a new continuum approach to consider burning wetness, temperature, and heat.*

The authors have provided a very complete and thorough response to my questions. Some of my questions originated from not being in the plume modelling field and I appreciate the authors working hard to explain what is going on and educate me. I commend the authors on such a thorough response, especially with current events as they are.

Very nice clear explanation of the plume model. I fully agree with the authors that simple models offer better insight in many situations because they aren't a black box.

Based on the additional explanation that the authors have provided on their formulation of the MLR I think their rationale makes sense. The added text is very helpful to readers such as myself to understand what each MLR is doing.

I appreciate the description of how the reanalysis has limitations in regions like the Andes, or when cloud cover is persistent. The authors' have clearly done a lot of work to make sure they understand what is going into making MERRA-2 work. I think the inclusion of the MERRA-2 data will be interesting to the NASA modelling community since it appears to work less well than the regression model. Modelling biomass burning is a known issue with MERRA-2 [Buchard et al., 2015] and evaluations such as this are important.

Overall I find the paper clear and improved and recommend acceptance.

Thank you very much for all of your support and deep insights throughout this entire process. We appreciate the fact that with your help, significant improvements have been made.

934: "Has difficulty in reproducing"?

This has been improved in response to a recommendation from the new reviewer. The new sentence is now:

*The present generation of models has difficulty to reproduce the actual vertical distribution of atmospheric aerosols when addressing cases that do not tend to have a combination of a highly energetic fire source, a relatively dry atmosphere, and a relatively optically thin smoke column emitted by the fire.*

1000/1023: "specifically"

This part has been corrected.

Pg 32 ln 065: Predictors in a regression model don't have to be orthogonal (https://www.statisticshowto.com/orthogonality/) or uncorrelated with each other. If the predictors are really strongly correlated it is a problem, but the authors are training and predicting on different data sets so it seems robust and I don't think they need to worry all that much. If the authors want to they could throw in a correlation matrix plot to show that the predictors aren't very strongly correlated, but I think that is unnecessary.

We agree with this observation. The strongest amount of correlation may be related to the overpass times, since the various measurements are all on the A-Train. But otherwise we do not observe any significant correlation, as demonstrated clearly in the new Figure 5.

00 Pg. 34 ln 143: 'which is produced more'

This has been clarified. Thank you!

05 Buchard, V., A. M. da Silva, P. R. Colarco, A. Darmenov, C. A. Randles, R. Govindaraju, O. Torres, J. Campbell, and R. Spurr (2015), Using the OMI aerosol index and absorption aerosol optical depth to evaluate the NASA MERRA Aerosol Reanalysis, Atmos. Chem. Phys., 15(10), 5743-5760, doi: 10.5194/acp-15-5743-2015.

[revised manuscript text omitted]

---

## Author Response (AR3)

Dear Editor,

Thank you very much for all of your careful and thoughtful suggestions throughout the process. We greatly appreciate your input and believe that it has helped produce a much better work overall. We have gone through the paper and made sure to edit the figures to make them more legible and easier to view. We also have fixed two very minor issues with regards to references. All changes are supplied in the track-changes mode file attached below.

We look forward to your positive response, and if you have any questions or suggestions, as always, please let me know.

Many thanks again for all of your time and assistance.

Kind Regards,

Jason
* * *
Editor Decision: Publish subject to minor revisions (review by editor) (07 Oct 2020) by Anja Schmidt
Comments to the Author:
Dear Authors,

I am pleased to let you know that I am happy to accept your manuscript subject to some additional minor revisions related to the Figures of your manuscript. You have done a good job addressing all reviewer comments but there are no axes labels on Figures 1, 2, and 6, and the font on a lot of your Figures appears too small in my pdf version of the manuscript to be legible. Please can you carefully check each figure, add labels and increase the font size etc and then upload a revised version of your manuscript? If you have any questions then please don't hesitate to get in touch with me.

Thank you for your submission to ACP.

Kind regards,
Anja Schmidt (as2737@cam.ac.uk)

[revised manuscript text omitted]